
# An autonomous and low-power instrument platform for monitoring water and solid discharges in mesoscale rivers

Guillaume Nord[1], Yoann Michielin[1], Romain Biron[1], Michel Esteves[1], Guilhem Freche[1], Thomas Geay[2], Alexandre Hauet[3], Cédric Legoût[1], Bernard Mercier[1]

[1]Univ. Grenoble Alpes, CNRS, IRD, Grenoble INP, IGE, 38000 Grenoble, France
[2]Univ. Grenoble Alpes, CNRS, Grenoble INP, GIPSA-lab, 38000 Grenoble, France
[3]Electricité de France, DTG, Grenoble, France

*Correspondence to*: G. Nord (guillaume.nord@univ-grenoble-alpes.fr)

**Abstract.** We present the development of the RIPLE platform designed for the monitoring at high temporal frequency (~ 10 min) of water discharge, solid fluxes (bedload and suspended load) and properties of fine particles (settling velocity) in mesoscale rivers. This platform responds to a request to continuously measure these variables in rivers using a single, centralized device, and in the most direct way possible. The platform integrates the following instruments: (i) for water discharge: water level and surface velocity radars, digital cameras, echo-sounder; (ii) for fine sediment load: turbidimeters,

automatic samplers including the SCAF (sediment settling velocity characterization device), (iii) for bedload: a hydrophone, (iv) for water quality: a conductivity probe and water sampling. As far as water discharge monitoring is concerned, priority has been given to non-intrusive instruments to improve the robustness of the system. All the instruments are driven by a data logger (Campbell® CR6), which stores locally the data and then upload them to a remote server every hour during the day using a 3G modem. SMS (Short Message Service) alerts can be sent depending on scheduled conditions (e.g. low battery

voltage, water level threshold, all samples of the automatic sampler collected). The platform has been designed to be as autonomous as possible: it is powered by a battery supplied by a solar panel. Limiting the power consumption of the platform was one of the main technical challenges, because of the quantity of instruments integrated. A simple 100 W solar panel is sufficient to power up the entire platform, even during winter or low insulation conditions. The RIPLE platform has been designed to facilitate its use and maintenance. A user-friendly interface has been developed enabling to visualize the data

collected by the platform from an internet connection. It is also possible to remotely configure the platform within this interface, for example to modify water sampling thresholds or alert thresholds. Finally, the platform is relatively easy to move from one site to another site because its installation requires little civil engineering. To date, RIPLE has been tested on two rivers of the Alps in France: the Romanche river in Bourg d'Oisans (45.1158°N, 6.0134°E, 710 m elevation) from September, 2016 to July, 2018 and the Galabre river in La Robine sur Galabre (44.1586°N, 6.2360°E, 680 m elevation) since October, 2018,

demonstrating the proper functioning of the platform.





## 1 Introduction

Sediment transport has an impact on the ecological status of water bodies, the morphological dynamics of the river, the stability of river banks and structures, as well as many human activities such as energy production and drinking water supply (Renwick et al., 2005; Lee and Foster, 2013). Two modes of transport are classically considered: bedload and suspended load (Julien, 1995). Bedload consists of coarse particles transported by sliding, rolling or saltation on the bottom of the river. Suspended load refers to the transport of fine particles through the turbulent field, within the water column. The river sediment material is generally a mixture of coarse and fine particles that can interact together in relation to hydrology (the succession of floods and low flows), vegetation and human activities (material extraction, riverbed modification, presence of dams and weirs) (Corenblit et al., 2007). The transfer of coarse and fine particles downstream can lead to progressive siltation of structures, such as hydroelectric power plants (Morris and Fan, 1998; Walling and Fang, 2003), or to degradation of river quality by clogging the bottom or by direct attack on the respiratory organs of fish species (Owens et al., 2005). Alternatively, high-energy rivers with insufficient supply of sediment load can lead to erosion of the main channel (Frings et al., 2014). Suspended sediments are also a privileged vector for the transport of nutrients (C, N, P) or contaminants (pesticides, metals, organic products, microorganisms). Finally, the increase in turbidity leads to significant over costs for water treatment when it is used as drinking water. Overall, there is a need for systems able to monitor water and solid discharges in rivers, and these systems should take into account both fine and coarse sediments.

High frequency monitoring of water and solid discharges is also a key element for the scientific community interested in the functioning of the Critical Zone (earth's near surface) in response to global changes (climatic change, land use and land cover changes). The understanding of the Critical Zone requires a holistic approach involving the monitoring of a combination of variables such as hydrological, chemical and biological variables (Brantley et al., 2016; Gaillardet et al, 2018) in different compartments of the catchment, i.e. soil, groundwater, river, surface-atmosphere interface, vegetation. One of the main issues of the people working on the Critical Zone is the determination of mass balance of water and associated mater and energy balance. The fluxes exported through the river system are of primary importance as the river collects a large part of the surface and subsurface flows within the catchment. There has been a recent attempt by the French Critical Zone community (Gaillardet et al, 2018) to develop a list of variables to be monitored continuously in each of the compartments of the Critical Zone. For rivers, this list includes variables such as flow discharge, electrical conductivity, temperature, turbidity, suspended sediment concentration (SSC), chemical and isotopic composition of the water.

The suspended sediment flux is usually obtained by multiplying the water discharge by the SSC, typically expressed in g/l of the mixture of water and suspended sediment. High-frequency SSC monitoring is required for reliable estimate of the





suspended sediment flux. Nevertheless, a reliable and easy method to obtain a direct, continuous SSC measurement is not currently available. Alternatively, a proxy of the SSC, which can be easily monitored continuously and related to SSC, is employed. The most commonly used proxy to date is turbidity (Gray and Gartner, 2009; Rasmussen et al., 2009; Navratil et al., 2011). In turbulent rivers, it is assumed that the SSC is relatively homogeneous within the cross section and therefore that

a point measurement of turbidity from the river bank is acceptable. This is easily the case in rivers where the slope of the bed is greater than 0.1% (Mano, 2008.; Navratil et al., 2011) and for silt size particles or finer. This assumption is questionable for sand-sized particles (Camenen et al., 2019). However, there is no reference method for sand-sized particles to date. The turbidity-SSC rating curve is established from direct measurements of SSC that are performed through automatic water sampling. Samples collected at regular intervals or when thresholds are exceeded (e. g. water level, turbidity) are returned to

the laboratory and analysed to measure the SSC using the filtration method (for low SSC, typically < 2 g/l) or the total evaporation method (for higher SSC) (Navratil et al., 2011; Nord et al., 2014). A global turbidity-SSC rating curve can be established using all direct SSC measurements performed at a station and allow the turbidity time series to be converted into the SSC time series. However, in headwater catchments where sediment sources can vary rapidly during a flood, it is strongly recommended to perform a specific calibration for each flood to reduce measurement uncertainty (Navratil et al., 2011). Indeed,

turbidity is very sensitive to particle size and also to the shape and colour of the particles.

High-frequency bedload sampling is required for reliable estimate of the bedload flux since bedload transport is a very dynamic process and can even be discontinuous through the occurrence of intermittent pulses (Aigner et al., 2017). However bedload samplings are very challenging to be performed continuously over long periods. Alternatively, hydraulically-based equations

are mostly used to compute bedload sediment budget (e.g. Recking et al., 2012) but their reliability is sometimes questionable (Gray and Simões, 2008). In the last decades, several proxies were used to get a continuous monitoring of bedload processes with surrogate methods (Gray et al., 2010). Among these methods, three of them do not need the installation of heavy, dedicated structures in the stream flow: (i) acoustic Doppler current profiler (aDcp), (ii) passive acoustic and (iii) seismic. aDcp can be used to measure an apparent bedload velocity which is related to bedload fluxes (Rennie et al., 2002, 2017). However, the

deployment of aDcp operating from the surface is not appropriate in the case of rivers with steep slope (typically > 1%) because the presence of waves at the surface of the water hinders the measurement and floating objects transported during floods may damage the device. Passive acoustic and seismic monitoring consist in recording the noises that are naturally generated by bedload transport when impacting the riverbed (Thorne and Foden, 1988; Tsai et al., 2012). Passive acoustic monitoring is achieved with hydrophones that sense the acoustic waves propagating under water. The acoustic power of bedload sounds has

been related to bedload fluxes by using site-specific calibration curves (Marineau et al., 2016; Geay et al., 2017a). Seismic monitoring is achieved with geophones or seismometers which record the seismic waves propagating in the soil layer (surface waves). Similarly, the power of bedload seismic noises has been related to bedload fluxes in laboratory experiments (Gimbert et al., 2018) and partially in field-experiments (Roth et al., 2016). Acoustic and seismic monitoring provide continuous proxies that are related to bedload fluxes but these proxies are also dependent on additional bedload parameters (i.e. grain size



distribution, bedload kinematics) and on processes related to wave propagation (Geay et al., 2017b). Up to date, whatever the indirect method used, calibration efforts with direct bedload samplings are needed to elaborate rating curves and to finally provide a continuous monitoring of bedload fluxes.

Whether it is for estimating suspended sediment fluxes, dissolved matter fluxes, nutrient or contaminant fluxes associated with fine particles, knowledge of water discharge is essential. The monitoring of water discharge is not easy, especially in mountainous rivers where flow discharge can vary of several orders of magnitude in a few hours (Borga et al., 2014) and solid discharges can demonstrate a very dynamic behavior over time, even more impulsive than the flow itself, resulting in significant changes in the morphology of the river. Therefore monitoring discharge deserves a special attention. The
conventional monitoring of flow discharge involves the measurement of a primary variable such as water depth or water level. A calibration curve, so-called stage-discharge rating curve, is usually established based on information such as gauging (punctual discharge measurements performed using different techniques such as slug injection of a tracer in solution, current meter, aDcp, handheld surface velocity radar gun) or hydraulic modelling to convert the water level time series into flow discharge time series (World Meteorological Organization 2010; Tomkins, 2012). This work typically involves significant
human and financial effort. It is indeed very demanding to mobilize technical work force to carry out gauging. Moreover, the mobilization of work force during floods is not without risk for the operators. It is also not so frequent to be present in the field during the biggest hydrological events. Finally, when a morphogenic flood occurs, the changes of the river geometry implies to start over the calibration curve. Thus, new methods are required to provide more direct access to the water discharge (without using the stage-dicharge rating curve) and therefore reduce the field work related to gauging and bathymetry surveys. These
methods involve the use of other variables in addition to the water level. The first variable that has been added to hydrometric stations is velocity. During the last decade, the introduction of fixed flow velocity monitoring systems (transit time ultrasonic flow meters, acoustic Doppler current profiler horizontal (H-aDcp) and vertical (V-aDcp), surface velocity radars) into hydrometric stations equipment has gradually become more widespread (Costa et al., 2006; Levesque and Oberg, 2012; Nord et al., 2014; Thollet et al., 2017). These systems all provide a more or less local monitoring of flow velocity. More recently,
systems using cameras and the Large Scale Particle Imagery Velocity (LSPIV) method have been developed and deployed in the field to estimate the surface velocity field (Hauet et al., 2008; Leduc et al., 2018; Le Coz et al., 2010 ; Muste et al., 2008; Stumpf et al., 2016). These systems provide field data of velocity on a stretch of river. Whether it is fixed flow velocity monitoring systems or LSPIV systems, there is no direct access to the mean channel velocity which is the variable of interest to estimate directly the flow discharge by multiplying the mean channel velocity by the cross-sectional flow area. Thus the
development of innovative methods is necessary to estimate the mean channel velocity from surrogate monitoring. The first attempts concern the index velocity method but this approach is still largely based on the use of gauging (Levesque and Oberg, 2012; Morlock et al., 2002). Continuous monitoring of the cross-sectional flow area by a direct method is not possible to date because of the lack of dedicated technology. Alternatively, a relation between stage and channel area, called a 'stage-area rating' is established based on topographic surveys, which are repeated typically once a year or after each morphogenic flood.


Changes in the water level-velocity relationship make it possible to detect changes in the bathymetry and the need to survey new bathymetry of the section (Thollet et al., 2017).

Therefore, being able to carry out high frequency monitoring of water and solid discharges in any river and at any site of
convenience is a high expectation for both operational and research applications. This should make it possible to extend the monitoring to isolated sites that have not yet been gauged, in environments that may be remote, difficult to access, under potentially extreme conditions (humidity, temperature, wind, radiation) or subject to destructive phenomenon (cyclone, floods, vandalism, etc.). The availability of power, continuous data transmission and possible remote configuration remains a challenge in these environments. There have been studies that have developed autonomous monitoring platforms in hostile
environments to remain unattended for several months or years (Musko et al., 2009; Bhatti and Ridley, 2014; Clauer et al., 2014; Peters et al., 2014; Morschhauser et al., 2017) but little or no for monitoring of water and solid (both fine and coarse particles) discharges in rivers (Mueller et al., 2013; Navratil et al., 2011; Comiti et al., 2014; Griffiths et al., 2014). This study presents the development of a multi-instrumental platform called RIPLE (River Platform for monitoring Erosion). The different variables measured relate to the flow discharge, water quality, fluxes of fine and coarse sediment and optionally the properties
of suspended fine sediments (settling velocity). A user interface is also presented that enables data visualization and remote configuration of the platform. A case study is described to validate the platform's operation.

## 2 Design consideration

The development of RIPLE is preferably aimed at mesoscale rivers, with potentially high SSC values (peak values typically between 1 and 100 g/l), flow discharge ranging over several orders of magnitude in a short period of time (from a few dozen
20    l/s to several hundred of m³/s) and bedload particles composed of a variety of grains from sand to cobbles or even boulders. However, it should be noted that the proposed instrumentation could be applied directly or after some modifications to lowland rivers and/or larger rivers. The choice was made to integrate into the platform instruments with a level of development that allows them to be easily assembled and to include recently developed instruments in the laboratory when company equipment do not yet exist. Priority was given to non-intrusive instruments because of their robustness. Indeed, floods may produce
important damage or destructions to any device present in the water. However, if non-intrusive instruments are commonly available for hydrometric applications, they do not really exist for the monitoring of sediment fluxes in this type of environment. The minimum set of variables to be integrated in the platform is as follows for: (i) hydrometry: surface velocity and water level; (ii) water quality: water conductivity, temperature and water sampling; (iii) suspended sediment: turbidity and SSC via water sampling; (iv) bedload: acoustic power and elevation of the river bed. A staff gauge and a control camera are
also required to establish a local elevation reference frame and check the general state of the platform. All instruments should be controlled by a central control system.

The platform should allow interaction between the different measurements (enabling triggering conditions to be activated and easily parametrized), data storage, remote data transfer, sending alarms, remote and real-time management. In addition, the platform should be autonomous in energy and easily movable from one site to another. The different functions taken into account in the development of this platform are shown in the diagram of Figure 1.

The sections presented in the following correspond to the sequential tasks that have been performed during the development of the platform: 1) the definition of the architecture and the choice of the acquisition and control system; 2) the development of an integrated solution: definition, development, test and formalization of protocols: power supply, sensors electronics, data storage, remote data transfer, enclosure/mechanics, and control software; 3) the development of a software interface allowing remote archiving and visualization of data and maintenance of the platform by sending new programs to modify the operation

of the control system and the interaction between measurements; 4) the deployment of the platform in the field for test and validation.

## 3 System design

### 3.1 Control system

The instruments integrated in RIPLE are listed in Table 1. They are sorted according to the type of measurement they provide:

hydrometry, water quality, suspended load, and bedload. The name of the variable, the physical principle of the measurement, the name of the model, the name of the manufacturer and the integration status in RIPLE are given for each instrument. A detailed description of each instrument is given in Section 3.3. All instruments are produced and marketed by manufacturers with the exception of the optic fiber turbidimeter (so-called "capteur marseillais") for high turbidity, and PASS (automatic water sampling) which are developed in public research laboratories and individually manufactured at the moment. The SCAF

(sediment settling velocity) has recently been marketed by the Aqualabo company. Two instruments, i.e. PASS (automatic water sampling) and SCAF, are now integrated in RIPLE but were not included in the case study presented in Section 6. They therefore appear as optional in Table 1.

The devices integrated in RIPLE are all controlled by the Campbell CR6 data logger (Figure 2). This data logger was chosen because of its robustness in isolated sites and under difficult environmental conditions (humidity, temperature), its flexibility

in terms of ports (universal ports), Ethernet port, Secure Digital (SD) memory card slot, large number of possible expansion modules (CDM-A108: analog expansion module, SDM-SIO1A: RS-232 RS-485 RS-422 expansion modules) enabling to control the large number of instruments integrated. This control system has a wireless connection option, allowing to remote control the system from the car or from a shelter during bad weather. The type of communication with each instrument is shown in Figure 2. The other technical elements necessary for the operation of the platform are a solar panel, a solar regulator,

a battery and a modem. The electrical diagram of RIPLE is presented in the Appendix A.

The sampling period of the platform is 10 minutes which is a good compromise between saving energy and obtaining a good description of water and sediment fluxes temporal variability in mesoscale rivers (Navratil et al., 2011). During the day, data

is streamed every hour via a 3G/GPRS modem to the server located in the laboratory in Grenoble and a digital control image is also transmitted. The modem is switched off at night to limit power consumption. For sites where a 3G connection would not be available, it is possible to switch to a 2.5G connection. It is still acceptable for ASCII data but it should be kept in mind that few images can be transmitted in this case. Alternatively, a satellite connection could be considered taking into account the higher subscription cost.

## 3.2 Power supply

RIPLE is powered by a combination of battery and photovoltaic panel to make it autonomous in energy and installable in a wide range of site, even if there is no wired power grid. The power of the solar panel must be checked for each installation according to the sunshine on the site (latitude and elevation of the site, orientation and angle of the panel, presence of masks). In Table 2, the detail of the power balance is displayed for the case study presented in Section 6. The total consumption of the platform was estimated at 7758 mA.h/day or 97.7 W.h/day. This calculation takes into account a 10 minutes sampling period for most of the instruments except the cameras which are switched on only every hour. Furthermore, the calculation is made with a sampling of 24 bottles during a single day. The data logger, which operates continuously, has the largest energy consumption, then comes the automatic sampler. Considering an autonomy of the battery of 5 days, which means a period of 5 days without any solar input, and a battery level that should not fall below 25% of its capacity, we obtain a capacity of 51.7 A.h. Considering an additional safety factor of 65%, the minimum battery capacity is 80 A.h. The selected battery model is a Sonnenschein Dryfit Solar Block SB 12 / 100A.h. The Photovoltaic Geographical Information System (PVGIS) proposed by the European Commission (http://re.jrc.ec.europa.eu/pvgis/) was exploited to define the power of the photovoltaic panel to be used in combination with the battery of 100 A.h. The results showed that a photovoltaic panel of 100 W was sufficient to comply with the criteria of charge of the battery (at least 25% of its capacity), even at the heart of winter, when the radiation is at its lowest. A photovoltaic panel Sunmodule SW 100 poly RIB (100 W) and a solar regulator STECA PR1010 10A/12-24V were therefore selected.

## 3.3 Sensors

The different instruments integrated in the RIPLE platform are presented in Table 1 and Figure 2. This section aims to provide more detailed information on each of the sensors and equipment.

### 3.3.1 Control camera

The control camera acts as a webcam, allowing to remotely visualize the instruments and the river. The camera can help to make a remote diagnosis in case of malfunctioning of the platform. The selected camera is an AXIS P1427-E based on various criteria such as image quality, zoom and focus, power consumption, robustness to the environment and transfer in real time of images. The maximal image resolution of the camera is 5 Mega Pixels (MP). Tests in the lab have shown that the image resolution should be at least 3 MP to allow details to be seen by zooming in the image. The minimum–maximum focal lengths

of the zoom lens are 2.8–9.8 mm. The lens has a fixed aperture of f/1.6. The horizontal angle of view ranges between 27° and 92°. A wide horizontal angle of view (at least 80° to 100°) is required to get an overview of the site and of all the sensors. It means that the entire section of the river on which the platform is located and the adjacent river reach (about ten meters upstream and downstream) should be visible in the image. This makes it possible to understand the hydraulic behaviour of the

river reach under all water level conditions. The control camera is usually installed at the opposite river bank to the staff gauge in order to see both the staff gauge and most of the sensors. The consumption of the camera is 5 W and it can be turned off while it is not taking pictures. The camera supports for HTTP, UDP and FTP protocols. The camera is connected to a Power over Ethernet (PoE) injector, which in turn is connected to the data logger via an Ethernet cable (see Appendix A). The PoE injector is connected to the power supply (12 V). The data logger controls a relay that supplies power to the PoE injector. In

terms of protection against solids and liquids, the enclosure of the camera complies with the Ingress Protection IP66. The camera can operate from -30 °C to 50 °C and with  relative humidity between 10% and 100% (including condensation).

The triggering of the acquisition is programmed as follows: (i) the camera is started by the data logger via the electrical relay (typically once an hour) (ii) the camera takes a picture at the end of a 2 minutes heating period (iii) the camera sends the picture to a FTP address via the data logger and also locally records a copy of the image on its 64 GB SD memory card (iv) the camera

is switched off via the relay.

Ideally it would have been possible to zoom in and even orient the camera remotely. However, these specifications were not retained since they would have had a strong impact on the power consumption of the platform (motors within the camera, need to keep the modem on when driving the camera) and on the amount of data passing through the network (limited by the GPRS subscription). As a result, the camera maintains fixed position, focal length and focus.

It is necessary to regularly retrieve the images that are stored directly on the SD card of the control camera by connecting to the camera from a Personal Computer (PC) using the Ethernet link from the RIPLE cabinet during the field visits. This allows to retrieve all the control images for subsequently archiving because it is possible for some images to be poorly or not at all remotely transferred by the data logger due to the variable quality of the 3G connection.

**3.3.2 Large Scale Particle Image Velocimetry digital camera**

This camera is dedicated to the Large Scale Particle Image Velocimetry (LSPIV) analysis, an optical technique for measuring surface velocity fields from image processing algorithms, analyzing the movement of natural tracers (leaves, floating branches, turbulent eddies) present on the water surface using a video of the river. A transect of surface velocity along the cross section of the river is extracted and converted to a transect of depth averaged velocity over the vertical using a coefficient that

commonly ranges between 0.75 and 0.85 (Hauet et al., 2008; Le Coz et al., 2010; Welber et al., 2016). Afterwards, knowing the cross section geometry, discharge is calculated by integrating depth averaged velocities along the cross section with an accuracy of approximately 20% (Welber et al., 2016).



The major advantages of this technique are the non-intrusive aspect (sensor out of water) and the automation of the acquisition which allows to obtain surface velocity fields without any manpower on site. There is no risk for operators and no risk of missing the peaks of the floods, with the exception of night time and technical problems. In this study, the LSPIV technique is not used to obtain continuous time series of discharge due to the limitations of the method (see below) and our inability to

monitor the bathymetry of the section continuously. Alternatively, this technique provides a set of "automatic" discharge measurements that makes easier and faster the building of a stage-discharge relationship. The main drawbacks of the technique are as follows: (i) a manual selection of the video sequences is necessary because some videos are not usable (e.g. lack of brightness, sun reflections, presence of dirt or water drops on the lens); (ii) the LSPIV processing steps are relatively time consuming and require a specific expertise. These steps include a possible correction (depending on the focal length used) of

image distortion, orthorectification of images (transformation of the image from fixed objects whose exact GPS location is known) in order to have the same scale at each point of the images, calculation of surface velocity and flow discharge. These steps are performed using the Fudaaa-LSPIV software, a free software available online (https://forge.irstea.fr/projects/fudaa-lspiv/files) that uses the Fudaa libraries released under the GPL licence; (iii) the method is not applicable during the night.

The selected camera is an AXIS P1435-LE based on various criteria such as image resolution, acquisition frequency, zoom

and focus, power consumption, robustness to the environment, possibility of recording on SD card and presence of an integrated IR projector for tests at night-time. The camera has an adjustable resolution from 160 * 90 to 1920 * 1080 (2 MP). Preliminary tests have shown that a minimum resolution of 600 * 400 is required for rivers with widths less than 30 m but resolution can be increased if needed. Resolution is more important for the precise positioning of the bitter points than for the visualization of the tracers. There is a link between resolution and focal length to be taken into account. The minimum–

maximum focal lengths of the zoom lens are 3–22 mm. The aperture is f/1.4 for focal lengths from 3 to 10.5 mm and f/1.85 for focal lengths from 10 to 22 mm. The horizontal angle of view ranges between 18° and 95°. This wide range of angle of view makes it possible to adapt to many sites. For positioning the camera and adjusting the angle of view, the following conditions must be respected (Le Boursicaud et al., 2016): (i) both river banks have to be visible in the image so that a complete cross-section is monitored; (ii) fixed and permanent markers (tree trunk, boulder, bridge pier, etc.) should be present within

the images, ideally placed on both sides of the flow; (iii) the targeted cross-section should be close to the cross section that includes the water level sensor and the staff gauge to have an accurate estimation of the wetted cross section. Furthermore, to optimize the results of the LSPIV technique, it is suitable to focus on an area where surface flow is as regular as possible; favour a stretch of river with a hard and stable bottom and avoid as much as possible solar reflections, scintillations and cast shadows. To meet all these criteria, it is preferable to install the camera at a significant height above the river. This will also

prevent the use of a low focal length that causes high distortion with this type of lens and is difficult to correct later.

The frame rate is up to 50/60 frames per second (fps) in all resolution. A minimum frame rate of 25 frames per second is used in this study. It is important that the interval between two images be constant and accurate. The consumption of the camera is 5 W and it can be turned off while it is not taking pictures. The camera is connected to a PoE (Power over Ethernet) injector (see Appendix A). The PoE injector is connected to the power supply. The data logger controls a relay that supplies power to

the PoE injector. In terms of protection against solids and liquids, the enclosure of the camera complies with the Ingress Protection IP66. The camera can operate from -30 °C to 55 °C and with a relative humidity between 10% and 100% (including condensation). The duration of the video sequences is 10 s. It is a compromise between, on the one hand, the minimum number of image pairs required to make a robust calculation and to average turbulent velocities and, on the other hand, the amount of

data to be stored. The video sequences can be recorded on a SD memory card. A 10-second film at a resolution of 2 MP and an acquisition rate of 25 fps without further compression typically generates a 30 Mega Bytes (MB) file. A 64 Giga Bytes (GB) SD memory card is therefore used in this application. For night measurements, an integrated Infra-Red (IR) projector based on highly efficient Light Emitting Diodes (LEDs) with adjustable intensity and angle of illumination is available. However, primarily tests have shown that the projector is not powerful enough to illuminate up to the water surface.

The triggering of the acquisition is programmed as follows: (i) the camera is started by the data logger via the relay for 2 minutes at regular intervals (typically every 30 minutes) when triggering conditions of turbidity and water levels are overcome, i.e. during flood (ii) the camera takes a short video (10 seconds) of the river and records the acquisition directly on its 64 GB SD memory card (iii) the camera is switched off via the relay.

The video sequences stored in the LSPIV camera are retrieved regularly during field visits by connecting to the camera from

a PC using the Ethernet link from the RIPLE cabinet. The LSPIV processing steps will be executed back in the laboratory in Grenoble. It is important not to change the position, angle of view and focus of the camera in order to easily reproduce the LSPIV processing chain. Finally, it was chosen not to use the same camera for the control and for the LSPIV, even if this solution had initially been considered, because the installation constraints specific to each of them are generally very different.

### 3.3.3 Surface velocity radar

The surface velocity radar model  RG-30 from Sommer company was selected in this study. It was designed for non-contact measurement of the surface flow velocity of river and channels. The sensor is mounted above the river, usually installed on bridges or river banks using extension arms (Figure 4). The radar sensor requires a low maintenance operation over many years.

The sensor emits a 24 GHz microwave beam (K-band) towards the water surface inclined by an angle of 58 ° from the vertical

axis. The radar sensor has an opening angle of 12°, hence, the projected area over the water surface of the river is an ellipse. The measurement of the flow velocity is based on the principle of the Doppler frequency shift. The reflected electromagnetic wave from the water surface is received by the antenna, analyzed and converted into surface water velocity. Every measurement is space averaged over the elliptical area targeted. The size of the targeted area, which depends on the distance from the sensor to the reflecting water surface, increases as water level decreases. The velocities appearing in this area have a

specific distribution depending on the flow conditions. The velocity distribution is determined with a digital signal processor via spectral analysis and the dominant velocity in the measurement area is calculated (internal processing developed by the Sommer company and not available to the user). Spectra can be output using the RQCommander software (Sommer company) by connecting directly a PC to the RG30 using a serial connection (RS485) with no connection to the data logger. These spectra

cannot be sent to the data logger, they can therefore only be used to evaluate the quality of the measurements during visits to the site. The quality of radar velocity measurements depends on flow surface patterns. As the flow surface gets exposed to external disturbance, caused by wind or rain, the system measurements become less reliable. Similarly, when no surface patterns are present on the surface, the measurement is biased (Welber et al., 2016). Velocity measurements are possible, if the wave height exceeds 3 mm, higher waves improve the reproducibility of the measurement. In the same way, best measurements are obtained when water is turbid.

The enclosure of the RG-30 complies with the Ingress Protection IP68. It has been designed to withstand exceptional floods with punctual immersion and can operate from -30 °C to 55 °C. The radar sensor can be mounted in a range from 0.5 to 30 m above the water surface or river bed. The radar sensor can either be mounted in or opposite to the flow direction. The view direction against the flow direction is recommended by the manufacturer. The measuring range is from 0.30 m/s to 15 m/s, the accuracy is +/- 0.02 m/s and the resolution is 1 mm/s. However, the experience shows that the signal can be noisy at low velocity, typically less than 0.5 m/s.

Every 10 minutes, the measurement is ordered by the data logger using a Serial Data Interface at 1200 baud (SDI12) protocol. Every measurement is time averaged over 30 records obtained in a time interval of 30 seconds. In addition to the velocity, a quality value of the measurement is provided and recorded. The radar is put on standby by the data logger between each measurement to limit power consumption.

### 3.3.4 Water level radar

The water level radar CRUZOE manufactured by Paratronic was selected in this study. It is designed for non-contact measurement of water level in rivers or channels. The sensor is mounted above the river, usually installed on bridges or river banks using extension arms (Figure 4). Flow areas that can be hydraulically disturbed by bridge piers must be avoided. The sensor should preferably be placed in the same section as the staff gauge.

The system emits a short microwave impulse (24.125 GHz, K band) towards the water surface and captures the reflected signal. The radar sensor has an opening angle of 12°. The radar antenna is both emitting and receiving. The sensor derives the time of flight of the impulse. Accounting for the velocity of the wave in the air and applying a correction with respect to the air temperature, the radar derives the distance separating the radar from the water surface. There is therefore a blind area of 0.15 m directly under the sensor.

The radar CRUZOE is easy to use. It does not require any parameter setting. Indeed, the "factory settings" allow its direct use in most cases. It is only necessary to convert the output value into an elevation in the staff gauge which is the absolute elevation reference of the station. Readings of the staff gauge during field visits or by mean of the images sent by the control camera allow to check the validity of the measurement and detect any possible changes in the geometry of the riverbed.

The enclosure of the CRUZOE radar complies with the Ingress Protection IP68. It has been designed to withstand exceptional floods with immersion of 100 days under 1 m and can operate from -20 °C to 60 °C. The radar sensor can be mounted in a range from 0.15 m to 30 m above the water surface or river bed. The accuracy is +/- 5 mm and the resolution is 1 mm.

This radar implements the JBUS protocol on an RS485 link. MODBUS instructions of the data logger allow communication with this radar. Every 10 minutes, the measurement is ordered by the data logger. Each logged value is the average of three measuring cycles, each lasting 4 s and separated by 5 s. During the measurement cycles, the instrument makes 16 measurements per second. In addition, the standard deviation of water level, the ambient air temperature, and four quality indicators of the distance measurement are recorded. The radar is put on standby by the data logger between each measurement to limit power consumption.

### 3.3.5 Conductivity and temperature probes

The Campbell Scientific 547 (CS547) probe with the A547 interface were selected for the measurement of electrical conductivity (EC) and temperature of water. The EC sensor consists of three stainless steel rings mounted in an epoxy tube. Resistance of water in the tube is measured by excitation of the center electrode with positive and negative voltage. Temperature is measured with a thermistor since the EC of a solution is highly dependent on the water temperature. Indeed, as the temperature of a sample increases, the viscosity of the sample decreases, resulting in increased ion mobility. As a result, the observed conductivity of the sample also increases, even if the ion concentration remains constant. To obtain comparable results, the measured values must be reported at a uniform reference temperature, generally 25° C. A simple method of correction of the effect of temperature on the EC measurement is applied assuming a linear relationship between temperature and EC.

The CS 547 probe is resistant to water and corrosion. It is easy to clean. The output signal is analogical (4-20 mA). The range of measurement for EC is from 0.005 to 7 mS/cm and the accuracy is +/- 10% of reading for 0.005 to 0.44 mS/cm and +/- 5% for 0.44 to 7.0 mS/cm. The range of measurement for temperature is from 0°C to 50°C and the accuracy is +/- 0.4 °C.

Every 10 minutes, the measurement is ordered by the data logger. Every measurement is time averaged over 30 records obtained in a time interval of 30 seconds. In addition to the average EC and average temperature of water, the min and max values of EC, the standard deviation of EC and the average value of raw EC (the value with no correction for temperature effect) are recorded.

### 3.3.6 Automatic river water samplers

Two types of automatic river water samplers were selected for this study: (i) the ISCO 3700 manufactured by Teledyne, and (ii) the PASS developed by IRD (Institut de Recherche pour le Développement) for low-cost applications with adaptable number and volume of sampling bottles. As shown in Figure 2, automatic water samplers are useful both for measuring SSC and for all subsequent analyses of dissolved and particulate phases (major ions, nutrients, contaminants, microorganisms, DNA...).

(i) The ISCO 3700 portable sampler is commonly used in hydrological, biogeochemical and suspended sediment studies. It can contain 24 wedge shaped, 1 liter polypropylene bottles or 24 cylindrical, 350 ml glass bottles. The ISCO 3700 allows to perform sequential or composite samples based on time or physical conditions that come from other sensors (e.g. water level, discharge or turbidity). In this study, only sequential samples are taken and the sampling is triggered with external impulses

coming from the data logger. The ISCO 3700 features a patented liquid detector. It is equipped with a peristaltic pump which delivers accurate, repeatable sample volumes time after time. The system includes an automatic compensation for changes in head heights and an automatic suction line rinsing to eliminate sample cross contamination. The pump maintains a suction velocity of 0.66 m/s recommended by ISO standard 5667-10 up to 4.5 m of vertical drop. The suction strainer is generally located relatively low in the river depth to sample any type of flood and any type of suspended particle.

Above some turbidity and water level thresholds, the values of which depend on the site and the season, the data logger will send an impulse to the sampler which will initiate the sampling of a river water sample. If both turbidity and water level thresholds continue to be exceeded and if a time interval is reached since the last sampling, the next sample will be collected. The time interval between two sampling depends mainly on the site and the season.

(ii) The PASS sampler is a more flexible alternative than the ISCO in terms of the number and shape of its containers. Any container in plastic, glass or other material with a top opening can be used. This system is controlled by a Campbell CR200 data logger that controls a pump and 4 stepper motors (2 in X and 2 in Y). This system was completely developed and tested in the M-TROPICS Critical Zone Observatory in Laos (Ribolzi et al, 2017). In this study, the same pump as the ISCO 3700 portable sampler was tested.

### 3.3.7 Turbidimeters

Two types of turbidimeters were selected for this study: (i) a standard instrument manufactured by MJK (the Susix sensor) for turbidity ranging from 0.001 to 9999 FNU/NTU (equivalent to suspended solids ranging from 0.001 to 400 g/l of $SiO_2$ according to the manufacturer) and (ii) a turbidimeter developed by IUSTI (Institut Universitaire des Systèmes Thermiques

Industriels) for the specific case of rivers with very high SSC (up to several hundreds of g/l of SSC).

(i) The SuSix sensor uses a multi-beam, pulsed infrared light system (wave length = 860 nm). The beam forming optics for multi-angle detection combined with a progressive algorithm using neural logic constitutes a reliable high quality measurement of turbidity in a single sensor according to the manufacturer. The turbidity measurement complies with ISO standard 7027.

The Susix sensor is equipped with a wiper to remove mineral and organic deposits from the optical cells. A SuSix Converter without display (10 - 30 V DC) on which certain parameters can be adjusted (unit, measuring range) is needed between the Susix sensor and the data logger. The standard RS-485 is used for the serial communication between the sensor and the converter with a proprietary protocol. An additional Display Unit for SuSix is installed and connected exceptionally for the



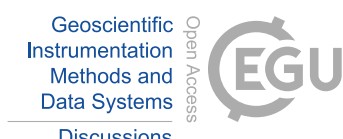

configuration of the system and the sensor. During normal operation, the converter without display allows the most economical use possible in terms of power consumption. The converter outputs a 4-20 mA signal to the data logger. The setting of the 4-20 mA output range depends on the turbidity range. In the case of a river with high SSC, the following values are used: 4mA = 0 FNU, 20mA = 9999 FNU. The SuSix sensor is constructed of stainless steel with chromium-dioxide coating and scratch

5    resistant sapphire lenses in a highly polished stainless steel sensor face. The accuracy is +/- 0,1% of reading.

The converter is started by the data logger via the relay every 10 minutes. Every measurement lasts for 30 seconds and records 30 values. In addition to the average turbidity, the min and max values of turbidity, the standard deviation of turbidity and raw value of turbidity in mV are recorded. Finally, the converter is switched off via the relay. The wiper for cleaning the sensor is activated when a 12V pulse is received, generated once a day by the data logger.

(ii) The turbidimeter developed by IUSTI, so-called "capteur marseillais", is a sensor made up of a bundle of optical fibers. This sensor which was initially designed for the Draix Observatory has shown great robustness, operating on site without maintenance since July 1994. Four sensors and their electronics are still in operation today. The operating principle of the "capteur marseillais" is described in detail by Bergougnoux (1995), Bergougnoux et al. (1998) and Bellino et al. (2001). The

sensor head is made of optical fibers with a diameter of 750 µm. There is an emitting fibre in the centre surrounded by two rings of receiving fibres. The first layer contains 6 receiving fibers, the second layer contains 12 receiving fibers. The sensor consists of a stainless steel head to be immersed in the river, a flexible black polyamide sheath to guarantee good mechanical protection of the optical fibers, a waterproof housing containing the LED and the two photodiodes to which the two families of receiving fibers are connected. An electronic box connected to the fiber optic sensor by 3 BNC connectors of 5 m cable

allows the amplification of the signals delivered by the two families of receiving fibers. This electronic box, powered by a 0-12V DC supply, is equipped with 2 analog outputs in voltage between 0 and 5V, compatible with the acquisition system. These devices are located in the main electronics box of the platform (Figure 3). The gains applied to 0-5V analog outputs must be adjusted in relation to the expected turbidity range. The ratio between the signals of the two receiver fibers layers gives a proxy of the turbidity. The accuracy of the turbidity measurement depends on the absorption capacity of the particles, it is about 3%

for suspended sediment concentration (SSC) between 1 and 40% in volume. This sensor is limited to the measurement of high turbidity values. In addition, it does not have a cleaning system (e.g. wiper or ultrasonic system). It was therefore decided to install it above the water surface for low flow conditions so that it would only be submerged during floods.

Every 10 minutes, the data logger starts up the power supply of the "capteur marseillais" via the relay. Every measurement lasts for 30 seconds and records 30 values. The recorded measurements are the average and standard deviation of the output

voltage of each receiver fibers layers as well as the average and standard deviation of the logarithm of the ratio between the signals of the two receiver fibers layers.



### 3.3.8 SCAF

The System Characterizing Aggregates and Flocs (SCAF) is an optical settling column composed of a series of 16 infrared emitters/receivers regularly spaced every centimeter (Wendling et al., 2015). The device allows to measure the temporal evolution of the vertical profile of optical absorbance during the quiescent settling of a suspension, immediately after its

5  sampling from the river. From the slopes of multiple isoabsorbance lines, settling velocity distributions (SVD) of suspended solids can be calculated, as well as an indicator of the propensity of particles to flocculate (Wendling et al. 2015). The SCAF is able to operate for a wide range of SSC (from one to several tens of g/l) and settling regimes (free, flocculated, and hindered settling regimes) as illustrated in Legout et al. (2018). The data produced by the SCAF are useful for the understanding and the modelling of the suspended sediment transport in rivers.

Each unit instrument was designed to be incorporated into sequential samplers. The SCAF was adapted to fit into typical 1 l wedge shaped polypropylene bottles used for ISCO 3700 automatic samplers. A round bottle in glass (0.20 m high and 0.035m in diameter) receiving the suspension sampled from the river (170 ml) and the associated memory card is housed inside the 1 l wedge shaped polypropylene bottle. The optical system is composed of 16 infrared ($\lambda$ =980 nm) emitters and 16 diametrically

15  opposed photo-sensors measuring at a frequency of 210 Hz. Currently, up to 8 units can be placed in a ISCO 3700 sampler dedicated to the SCAF device.

The SCAF measurements begin after a delay of 20 seconds following the impulses emitted by the data logger to the ISCO. The delay corresponds to the time of purge and pumping of the 170 ml of the suspension by the ISCO sampler. Measurements are acquired every 100 ms each value being the average of 10 measurements. The SCAF measurements typically last over 5

20  hours as shown in Legout et al. (2018). Each SCAF unit is working separately, waiting for the various impulses sequence of the data logger.

### 3.3.9 Echo sounder

The echo sounder Airmar SS510 was selected to be integrated into RIPLE platform. It allows to perform a continuous

25  measurement of the distance to the riverbed at one point of the cross section. As a result, changes in the geometry of the cross section related to erosion/deposition of sediment or bedload transport can be detected at high temporal resolution, especially during flood events when geomorphic processes may occur. As shown in Figure 2, the echo sounder provides information on the bathymetry of the cross section that is useful for both bedload and hydrometry. The Airmar SS510 sensor, featuring embedded micro-electronics, processes depth and temperature signals inside the sensor, transmits data via two separate

30  communication protocols. The first is a bi-directional interface compliant with the NMEA-0183 protocol and the second is a transmit-only interface with a proprietary protocol using RS-485, which is used in this study. The acoustic frequency used by the sensor is 235 kHz. The power output from the transmitter is 100 W. There are minimal side lobes for concentrated energy

on target. The beam width is 8° and the range of depth is between 0.4 and 200 m with an accuracy of +/-0.03 m for applications in mesoscale rivers (depth typically lower than 5 m) with the sensor tilt with angles of less than 25° from the vertical axis. The accuracy of temperature sensor is +/-0.05°C. The operating temperature ranges from -5°C to 60°C. The sensor is robust with stainless-steel housing and an acoustic window in urethane. The cylindrical shape and the relatively small size (diameter of

0.007m) of the sensor offers flexibility in types of mounting. An adjustable attachment piece was designed in this study to fix the echo sounder from a river bank or bridge pier. The mechanical integration of the echo sounder into the river is the main problem associated with this instrument as it is potentially exposed to bedload transport and there is not always a hard point available to fix it.

Every 10 minutes, the measurement is ordered by the data logger via a RS-422 serial communication. By default, the echo
sounder returns the measured depth and water temperature every second. Every measurement is time averaged over 30 records. The recorded measurements are the average, min, max and standard deviation of depth as well as the average, min, max and standard deviation of water temperature.

Additionally, the RS-485 interface available on the echo sounder allows to retrieve detailed information on each measurement made by the echo sounder. To do this, it is necessary to establish a connection from a PC to the echo sounder with a converter
from FTDI (Future Technology Devices International) USB to RS-485 and open a 921600 baud terminal to display this data.

### 3.3.10 Hydrophone

The deployment of hydrophones in the watercourse enables a continuous monitoring of the sounds naturally generated by bedload transport in the river (Marineau et al., 2016; Geay et al., 2017a). The hydrophone Colmar GP0190 interfaced with the SDA14 acoustic data recorder were selected in this study. The Colmar GP0190 is a preamplified omnidirectional hydrophone,
for application up to 170 kHz (working band: 5 -170.000 Hz). The hydrophone can work up to 1000 m depth. The body of the instrument is in stainless steel. The SDA14 acoustic data recorder was designed by the RTsys company for the acquisition of acoustic signals from passive or pre amplified hydrophones. It integrates four analog receivers, allowing recording four sound sources simultaneously. Its broadband analog inputs allow over 500 kHz (from 3 Hz to over 500 kHz) with a dynamic range greater than 100 dB guaranteeing efficient signal to noise ratio. The embedded digital signal processor allows high speed
acquisition, filtering, storage and pre-processing of the acoustic data. Its power consumption is between 600 mW to 2 W in active mode (i.e. during measurements) and less than 1 mW in sleep mode. The system is designed to operate in standalone mode or towed mode. The standalone mode is used in this study. Configuration of the SDA14 acoustic data recorder is possible by connecting a PC via Ethernet and using a web interface.

The SDA14 is controlled by the data logger via an RS-232 link and a 0-5 V output. The triggering of the acquisition is programmed as follows: (i) the data logger sends a 5V pulse to wake up the SDA14; (ii) the SDA14 starts and automatically launches an acquisition (manual mode): a record of 30 seconds at 156 Khz and with a 24 bits resolution is performed. For information, the parameters of duration, frequency and resolution of the acquisition can be modified in the data logger

acquisition program; (iii) data are stored in files with .wav format; (iv) a fast Fourier transform algorithm is operated by the SDA14 to compute the acoustic root mean square power in third-Octave bands; (v) the SDA transmits integrative spectrum data (RMS power in third-Octave bands, about twenty points of the spectrum) to the data logger via an RS-232 link, so that useful information on the operation of the hydrophone can then be tele-transmitted to the manager of the platform (it would

not be possible to tele-transmit the .wav files by GPRS or 3G as they are too heavy, i.e. 10 MB for 30 seconds of recording) (vi) the data logger puts the SDA14 in standby via the RS-232 link.

Concerning step (iii) on data storage, the SDA14 is equipped with a 128 GB SD memory card and a 2 Tera Bytes (TB) hard disk drive (HDD) (ext4 format). In order to limit the number of on/off times of the HDD (which would severely limit its

lifetime), acquisitions are first recorded on an SD memory card and once it is full, the HDD is turned on to empty the SD card. This is called the "hybrid" mode. There is an RS-232 command returning the available memory in the SD card and in the HDD. The data logger will send an alert when the hard disk is almost full. The SDA14, the SD memory card and the HDD are located in the main electronics box of the platform. The transfer time from the 2 TB HDD to a PC would be far too long to be done in the field. A second HDD was therefore purchased to allow rotations to be done: when a disk is almost full (the state of the

storage of the HDD is visible in the RIPLE interface), it is simply replaced by the second HDD previously emptied, and the transfer of the full HDD can be done in the laboratory.

The hydrophone is housed in a polyethylene tube next to the other immersed instruments (turbidimeter, conductivity and temperature probes) which are not switched on during hydrophone acquisitions.

### 3.4 Remote data transmission

The 2G/3G modem Erco&Gener GenPro 325e was selected in this study as it allows the data logger to upload data and images to the FTP server of the laboratory in Grenoble and also to send alert SMS (text) messages. 2G antennas are no longer maintained by access providers in France, so we opted for a modem that can use 3G to remotely transmit data. A SIM

(Subscriber Identity Module) card linked to an M2M (Machine to Machine) subscription is inserted into the modem to ensure its operation.

The solution of the private IP address was selected in this study. By making this choice, we accept the dependence on the Loggernet software of Campbell Scientific, which makes it easy to manage the data logger and set up automatic collections of

data. Having a private IP address avoids the possibility of being hacked by a "robot" circulating on the net, which could cause a very significant increase in expenses related to the GPRS subscription. In France, Internet service providers only provide dynamic IP addresses. The Loggernet software was configured to establish a connection with a station whose address is dynamic by accepting the possibility of losing temporarily the connection because of a change in IP address.

All the procedure for the data collection and data transfer to the remote FTP server is presented in Figure 7. The modem queries the Internet service provider to obtain a private IP address. The data logger is then connected to the 2G/3G network. The data logger contacts the Loggernet server via port 6786. The Loggernet software running on the Loggernet server recognizes the

data logger with its Pakbus number (225 for RIPLE), the connection between the data logger and the Loggernet server is then established. It is then possible for the manager of the platform to communicate remotely with the data logger. The Loggernet software can manage different stations, the distinction between different data loggers is made by a unique internal Pakbus address assigned to each station and not by IP address or domain name. In case of a change of IP address, a connection failure occurs until the data logger automatically sends the next tag to the Loggernet server and can communicate once again with it.

The IP address is changed on average every 24 hours and also when the modem starts up. It is unlikely that the change of IP address occurs within a few hours after the modem is started. As long as the connection is established, the data logger sends its data tables and control images to the FTP server of the laboratory using FTPClient instructions.

The advantages of the selected solution are (i) the possibility to use a classic GPRS modem (low energy consumption and low

cost), (ii) the management of data flow in case of transmission errors, (iii) the possibility to have a 2-way communication link. The shortcomings are (i) the dependency to the Loggernet package proposed by Cambpell Scientific, (ii) the likelyhood to lose temporarily the connection if the IP address is changed.

The CR6 turns the modem on during the day and off at night to limit the power consumption of the platform. Therefore, no

data or alerts can be transmitted during night.

### 3.5 Housing

RIPLE platform is organized in several parts: (i) the control block, (ii) submerged instruments and (iii) non-intrusive instruments.

(i) The control system (CR6, CDM-A108, 4 SDM-SIO1A), the power supply (battery, solar regulator, 4 relays, 2 PoE injectors), the instrument electronics boxes (SuSix turbidimeter interface, power supply of the "capteur marseillais", A547 interface of the conductivity probe, SDA14 card of the hydrophone) and the remote transmission module (modem, antenna) are grouped in an electrical box (dimensions H = 1.40 m, L = 0.80 m, D = 0.46 m) located on the the river bank at a height sufficient to avoid being flooded (Figure 3). Everything is grouped into a single element to facilitate the installation and

relocation of RIPLE to other sites. In addition, the ISCO automatic sampler is placed next to the electrical box and the solar panel is fixed on a mast or against the structure of a bridge.



(ii) Three main submerged instruments (conductivity probe, SuSix turbidimeter, hydrophone) are housed in polypropylene tubes, 3 m long and 0.10 m in diameter, fixed to the bank of the river, parallel to each other. These tubes are clamped between metal profiles at 3 points (top, middle and bottom of the tubes). The metal profiles are themselves anchored in the bank of the river (bedrock or large blocks) using threaded rods. The 3 polypropylene tubes are perpendicular to the direction of the flow

in the river (see Figure 4). The instruments are installed within the tubes using PVC pieces of the inner diameter of the tubes that are machined to allow the sensors to be inserted. These PVC pieces, connected to the top of the pipes by 4 mm diameter threaded rods, prevent the movement of the sensors inside the pipes, allow the sensors to be easily removed without human intervention in the river and allow to put the sensors back to the same location. The lower end of the tubes is at a level low enough to ensure that the instruments are submerged during low water periods.

In addition, the "capteur marseillais", which is composed of a waterproof box including the LED source and photodiodes and a 1.5 m long sheath including the optical fibers, are fixed to the outside of the polypropylene tubes using cable ties. The dimensions of the box and the length of the optical fibers mean that this instrument cannot be housed inside the polypropylene tubes and the optical fibers heads cannot be lowered to the lower end of the tubes. The measurement is made at a higher level in the water column than the SuSix turbidimeter. The flexible plastic tube that allows ISCO to collect water and suspended

sediment from the river is also attached to the polypropylene tubes using cable ties. The end of the flexible tube is positioned very close to the SuSix turbidimeter, at the same level in order to have a maximum correspondence between the two measurements. The other submerged instrument that is not installed within the polypropylene tubes is the echo sounder. This one has a specific support that has been designed to be fixed to a vertical wall (bridge pier for example) by adjusting the angle of orientation of the instrument with respect to the vertical. Finally, a staff gauge is installed in the cross section near the water

level radar.

(iii) The four non-intrusive instruments used for hydrometric purposes are fixed on masts for both cameras (control, LSPIV) and on extendible mounting brackets for both radars (water level, water surface velocity) as shown in Figure 4. The brackets are easily movable to facilitate radar maintenance. These devices (masts or mounting brackets) are attached to the structure of

a bridge for example. The velocity radar should preferably be placed in the center of the cross section, in the zone of highest velocities. The cameras must be located on the banks at a level high enough to see the full width of the river and part of the banks.

## 4 Control software

The data logger controlling program is written in CRBasic programming language, the proprietary format of Campbell

Scientific. As shown in Figure 5, there is a main program that reads a configuration file, initializes the instruments and controls two families of subprograms: those which are active every 10 minutes or every hour.

The subprograms that are active every 10 minutes control all instruments except the control camera. When the water level and turbidity conditions are exceeded and the time since the last sample exceeds a certain interval set by the user, the first automatic sampler is launched. A SCAF measurement is also started, preferably using a second automatic sampler dedicated exclusively to SCAF measurements. Similarly, when a water level condition is exceeded and the time since the last video shot exceeds a

certain interval set by the user, the LSPIV camera records a film sequence. The other subprograms that are called at an hourly frequency are only active during day time. This concerns the operation of the modem, image capture by the control camera, image and data transmission, SMS message sending in case of alerts (low battery voltage, full ISCO sampler, etc.).

RIPLE platform is interfaced with the rest of the world via the FTP server, which allows exchanges between the interface (see

section 5) and the platform as shown in Figure 6. The procedure for the operation of data storage and transfer can be illustrated by Figure 7.

Concerning specifically the control camera, we use the data logger as an FTP server, on which the camera places an image every hour. The data logger then transmits this image to the FTP server of the laboratory in Grenoble. Since the Loggernet

software is not able to automatically collect files other than data tables, the data logger must therefore perform PUSH on the FTP of the laboratory server to retrieve the control images (the data logger can be both server and FTP client). The advantage of this method is that a traditional GPRS modem can be used. A drawback is that the images have to pass through the data logger storage memory but this does not have much impact on performance and consumption of the control system.

## 5 User interface

A solution with a remote web server was chosen, i.e. it is the server in the lab that generates a web page from the collected data. The RIPLE user interface is developed in R using the Shiny package (JavaScript elements for web interfaces) and dygraphs (graphics). An executable file for this application has been generated to display the results on a dedicated web page to avoid having to install RStudio and to have access to the interface from any terminal equipped with an internet connection.

### 5.1 Data visualization

By default, the interface starts on the "Data visualization" menu, in which all the data transmitted by RIPLE platform can be seen (the display may take a little time due to the amount of data). It is possible to choose the type of time series to display:
- Fixed time series (default): these are 4 graphs displaying the more common data as shown in Figure 8, the water level being present in each of these graphs as a common reference. There is a first graph with water level and surface water velocity, a second graph with water level and turbidity, a third graph with water level and water temperature, a fourth graph with water

level and water electrical conductivity. These graphs gives an overview of how the station works for the basic variables.


- Customized time series (optional): there is a single graph on which it is possible to add two curves on each y axis, among all the variables transmitted by RIPLE platform as illustrated by Figure 9.

All the data that can be displayed is read from the Riple_DATA.txt file (Figure 6) that is located on the FTP server in Grenoble, i.e. the file uploaded by RIPLE platform.

For each type of time series, it is possible:

- To modify the time window to be displayed. There are four options: the last day, the last week, the last month or a manual selection of start and end dates.

- To download the displayed data in an ASCII file in the same format as the one originally produced by the CR6. It is possible

to select either only specific variables or all variables as in the file uploaded by RIPLE.

## 5.2 Control images

A second menu "Control images" allows to remotely view the RIPLE platform by displaying the control images that are stored on the FTP server in the laboratory. For example, it allows to visualize the hydraulic behaviour during floods and at low flows (see Figure 10). Only fully transmitted images are accessible by default. It is still possible to consult all the control images

later on, after having retrieved manually the control images during field visits. All control images are thus stored in an archive directory independent on the FTP server.

## 5.3 Supervision

The "Supervisions menu" allows to remotely control the proper functioning of the platform. The data displayed in this menu are read from the Riple_SAV.txt file that RIPLE uploads to the FTP (Figure 6). These are technical variables concerning the

20 control unit (reference identifiers, OS version, internal battery voltage, PakBus address), the name of the current program of the control unit, the status of the power supply and temperature in the electrical cabinet of RIPLE platform, the data collected (number of measurement, date and time of measurement, watchdog errors, skipped scans, error with the SDA14 card, status of the SD memory card and of the HDD of the hydrophone).

For example, it is possible to check the status of the power supply of the platform by looking at the battery voltage time series

(Figure 11). The temperature measured by the data logger, i.e. the temperature inside the cabinet, must also be controlled, especially in winter (be careful if air temperature is below -10 °C) and in summer (be careful if air temperature is above 45°) because the battery is also present in the electrical box. Based on these thresholds, a SMS alert is sent and the fan is triggered. In addition, if a "skippedScan error message" appears, it means that the measurement time of some instruments should be reduced to allows the data logger to perform all the measurements in one cycle.



## 5.4 Configuration

Some parameters of the RIPLE platform can be configured remotely, from the "Configuration menu" of the interface (see Figure 12). This menu is only used by RIPLE's main "administrators". To modify other variables, e.g. duration of a scan, measurement time on each instrument, it is necessary to modify the Campbell program of the data logger. It is also possible to

do it remotely but it is preferable that it remains exceptional.

## 6 Case study

To date, RIPLE platform has been tested on two rivers of the French Alps: the Romanche in Bourg d'Oisans (45.1158°N, 6.0134°E, elevation 710 m) from September 2016 to July 2018 and the Galabre in La Robine sur Galabre (44.1586°N, 6.2360°E, elevation 680 m) since October 2018. Photos of the installation sites are shown in Figure 13.

The first site corresponds to a large embanked river typical of anthropized alpine valleys: presence of dams upstream and dikes giving rise to a very rectilinear river. The width of the river is about 30 m, the depth is typically between 0.5 and 1 m at low flows. SSC typically changes between 0 and 10 g/l all over the year. The platform has benefited from the existing hydrometric station managed by the Electricité De France (EDF) company, including regular discharge measurements and a reference stage-discharge rating curve.

The second site corresponds to a more pristine river in the Southern Alps where sediment loads can be high (max SSC ~ 360 g/l) (Esteves et al., 2019), due to the presence of numerous active badland areas. The width of the river is about 10 m, the depth is typically 0.1-0.2 m at low flows. The station also benefited from a hydrosedimentary station located 2,5km upstream (Esteves et al., 2019), managed since 2007 by the IGE and belonging to the Draix-Bléone observatory and the research infrastructure OZCAR (Gaillardet et al., 2018).

During these two years of testing on two sites, the platform has worked properly, recording a large data set that will be of great interest for the understanding of sediment transport processes in alpine rivers. The use of RIPLE data is in progress. For example, a current work is being done to combine radar surface velocity measurements with LSPIV velocity measurements to estimate the mean channel velocity and identify the moments when the geometry of the river is significantly modified by deposition and erosion processes.

## 25    7 Conclusions and outlook

The characteristics of the presented platform dedicated to monitoring erosion in mesoscale rivers result from a fifteen-year expertise in hydrometric and sediment measurements within the IGE laboratory and more broadly within the research laboratories in the Grenoble and Lyon communities and local companies such as EDF. The platform has been designed to be applied preferably to rivers in mountainous areas, but much of the system is transferable to lowland rivers. Through this study,

emphasis was put on water discharge as this elementary variable is not trivial to monitor in situ. Further developments are

needed to improve this measurement, especially in rivers where geomorphological processes are very active. A future objective is to produce methods for accessing continuously and directly to discharge data, with a very limited use of gauging and human resources. In addition to the set of measurements provided by the presented platform, a major challenge for the next decade, will be the development of methods allowing to monitor continuously the bathymetry within the river cross section.

For fine sediment transport, the generalization of the use of the turbidimeter associated with automatic river water sampling for the calibration of turbidity-SSC relationships has allowed a fairly rapid extension of the monitoring of suspended sediment fluxes during the last decades. However, it is still necessary to improve the spatialization of this measurement within the river cross section, particularly in the case of coarse silt or sandy sized particles, which often do not have a homogeneous concentration profile within the water column but rather an increasing concentration profile with depth. It seems necessary to develop non-intrusive technologies for the measurement of suspended sediment fluxes to increase the robustness of the measurement and reduce in situ maintenance. Furthermore, the measurement of the physical characteristics of particles is important because it provides information on transport capacity and deposition processes, on the spatial origin of eroded particles and indicates the propensity of particles to transport adsorbed substances (nutrients, metals, organic products, micro-organisms, micro plastics, etc.). These measurements must be performed under conditions that most closely resemble those of the in situ environment to avoid subsequent flocculation/disaggregation processes.

Concerning coarse sediment transported by bedload, recent metrological developments make it possible to start considering continuous and high frequency monitoring of fluxes and physical characteristics of particles (size distribution) using proxies and inversion models. Passive acoustic and passive seismic methods are experimenting on-going developments. The first results point to a promising future, although the difficulty lies in validating these methods. Indeed, in situ sampling of transported coarse particles is difficult to carry out and cannot be automated. In this study, the choice was made to integrate passive acoustics technology. These measurements of bedload correspond to a strong demand from the scientific community and more generally from the society.

Finally, water quality is partially taken into account in the RIPLE platform through the measurement of electrical conductivity and water temperature and also through the automatic sampling of river water which allows, after a filtration step in the laboratory, to carry out any type of analysis from the filtered phase, so-called dissolved phase (chemical, microbiological, DNA, etc.). Automatic sampling makes it possible to collect water during flood periods or during low water conditions. The collected samples are also useful for performing analyses (chemical, microbiological, DNA, etc.) on the particulate phase.

In the end, we want to show that RIPLE platform is a unifying tool that contributes to multidisciplinary studies on understanding the functioning of the Critical Zone. This is the way this tool has been designed and will continue to evolve. RIPLE platform is in constant evolution: new innovative instruments are integrated when they have been validated and are in

a development phase that allows their integration. Recent examples of integration are the SCAF and the hydrophone. RIPLE is an autonomous and low-power instrument platform, which transmits real time data to a remote server and can be controlled remotely enabling to fully exploit its potential. The visualization software interface that has been developed allows an easy follow-up of all measured variables and a beginning of data quality control.

The CRITEX project has made it possible to purchase an aquatic drone, the FoRiver 1 manufactured by River Drone, which offers the opportunity to plan spatial campaigns of certain variables at "hot moments" (low water level, flood, hydraulic flushing). The drone can thus carry instruments on board to perform measurement campaigns of the same variables as those measured by RIPLE platform at other points in the cross section or at other points in the profile along the river at different moments.

*Author contribution*

Yoann Michielin developed and implemented the integration of the RIPLE platform and developed the visualization interface. Michel Esteves and Guillaume Nord supervised the work of Yoann Michielin. Thomas Geay contributed to the integration of the hydrophone into RIPLE. Cédric Legoût and Bernard Mercier contributed to the integration of SCAF into RIPLE. Alexandre Hauet contributed to the integration of the control camera and the LSPIV camera into RIPLE. Yoann Michielin and Guillaume Nord supervised the installation of RIPLE on the Romanche site, with the help of the IGE technical service. Romain Biron and Guillaume Nord supervised the installation of RIPLE on the Galabre site, with the help of the IGE technical service. Romain Biron and Guilhem Freche are responsible for the maintenance of RIPLE. Romain Biron supervises RIPLE on a daily basis. Guillaume Nord wrote the manuscript using as main basis the technical documentation of the platform written by Yoann Michielin. All the co-authors reviewed the document and contributed more specifically to certain sections.

*Competing interests*

The authors declare that they have no conflict of interest.

*Acknowledgments*

This work was supported by the EQUIPEX CRITEX project (grant no. ANR-11-EQPX-0011, PIs J. Gaillardet and L. Longuevergne). Yoann Michielin benefited from an engineer contract from the CRITEX project. The work of Thomas Geay was funded by a research program between Electricité de France (EDF) and Université de Grenoble (GIPSA-Lab / IGE). The development of this platform was carried out with the support of the technical service of the IGE. The deployment of RIPLE platform in the field was also supported by Labex OSUG@2020 (ANR10 LABX56) and the Institut National des Sciences de l'Univers (INSU/CNRS). Most of the instruments presented in this study are the property of the CNRS. They are part of the national park of instruments for the study of the Critical Zone set up as part of the CRITEX project (https://www.critex.fr/).


The authors thank Norbert Silvera for his help in integrating the PASS sampler into the RIPLE platform. The authors also thank the French research infrastructure OZCAR (Observatoires de la Zone Critique, Applications et Recherche), the Draix-Bléone observatory, EDF, the AD Isère Drac Romanche and the department of Alpes de Haute Provence for allowing RIPLE platform to be installed on the Romanche and Galabre rivers.

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





## Tables and Figures

**Table 1: Selection of instruments integrated in RIPLE paltform. The "ni" exponent in the column "Instrument name" means non-intrusive instruments.**

| Compartment | Variable | Physical principle of measurement | Instrument name | Manufacturer | Integration in RIPLE [1: yes, opt: optional] |
|---|---|---|---|---|---|
| hydrometry | Water level | Time of flight radar | cruzoe [ni] | Paratronic | 1 |
| | Surface water velocity | Doppler radar | RG30 [ni] | Sommer | 1 |
| | Surface Velocity field | Camera + LSPIV* analysis | P1435-LE [ni] | Axis | 1 |
| water quality | Conductivity and Temperature of water | Conductivity probe | CS547 | Campbell Scientific | 1 |
| | Properties of the dissolved phase | Automatic water sampling (24 x 1 liter polypropylene or 350 ml glass) | 3700 | Teledyne ISCO | 1 |
| | | Automatic water sampler (variable size and number of bottles) | PASS | IRD | opt |
| suspended sediment | Turbidity | Optic backscattering (standard) | susix | MJK | 1 |
| | High Turbidity (concentration 10 -> 500g/l) | Optic backscattering (laboratory development) | capteur marseillais | IUSTI Marseille | 1 |
| | Suspended sediment concentration | Automatic water sampling (24 x 1 liter polypropylene or 350 ml glass) | 3700 | Teledyne ISCO | 1 |
| | | Automatic water sampler (variable size, number and material of bottles) | PASS | IRD | opt |
| | Properties of the particulate phase | Automatic water sampling (24 x 1 liter polypropylene or 350 ml glass) | 3700 | Teledyne ISCO | 1 |
| | | Automatic water sampler (variable size, number and material of bottles) | PASS | IRD | opt |
| | Fall velocity | optical absorbance | SCAF ** bottles | IGE | opt |
| bedload | Acoustic power | Hydrophone (passive acoustic) | GP0190 | Colmar | 1 |
| | River bed elevation | Echo sounder (Acoustic time of flight) | echorange | Airmar | 1 |

* LSPIV : Large Scale Particle Image Velocimetry
** SCAF : Suspended aggregates and flocs caracterisation system

IUSTI is an acronym for Institut Universitaire des Systèmes Thermiques Industriels, IRD is an acronym for Institut de Recherche pour le Développement, IGE is an acronym for Institut des Géosciences de l'Environnement





**Table 2**: **Power balance of instruments integrated in RIPLE platform.**

| System | Number of measurements per day | Operating consumption [mA.h/day] | Standby consumption [mA.h/day] | Integration in current version of RIPLE [0: no, 1: yes] | Total daily consumption [mA.h/day] | | Total daily consumption [W.h/day] | |
|---|---|---|---|---|---|---|---|---|
| Campbell CR6 | x | 1560.0 | 780 | 1 | 2340.0 | | 29.5 | |
| Module CDMA-108 | x | 240.0 | 187.2 | 1 | 427.2 | | 5.4 | |
| Modem | 1 | 193.7 | 10 | 1 | 203.7 | | 2.6 | |
| Water level radar | 144 | 18.0 | 0 | 1 | 18.0 | | 0.2 | |
| Water velocity radar | 144 | 132.0 | 22.8 | 1 | 154.8 | | 2.0 | |
| LSPIV camera | 24 | 333.3 | 0 | 1 | 333.3 | | 4.2 | |
| Control camera | 24 | 333.3 | 0 | 1 | 333.3 | | 4.2 | |
| IQ Plus | 144 | 300.0 | 48 | 0 | 0.0 | | 0.0 | |
| Conductivity probe | 144 | 12.0 | 0 | 1 | 12.0 | | 0.2 | |
| S::CAN | 144 | 840.0 | 108 | 0 | 0.0 | | 0.0 | |
| Turbidimeter | 144 | 664.0 | 0 | 1 | 664.0 | | 8.4 | |
| Optic fiber turbidimeter | 144 | 540.0 | 0 | 1 | 540.0 | | 6.8 | |
| Aquascat | 0 | 100.0 | 1.9 | 0 | 0.0 | | 0.0 | |
| Water sampler | 0 | 1600.0 | 165.2 | 1 | 1765.2 | | 22.2 | |
| SCAF | 4 | 666.7 | 0 | 0 | 0.0 | | 0.0 | |
| Hydrophone | 144 | 400.0 | 86.4 | 1 | 486.4 | | 6.1 | |
| Echo sounder | 144 | 480.0 | 0 | 1 | 480.0 | | 6.0 | |
| **TOTAL** | | **8413.0** | **1409.5** | | **7758** mA.h/day | | **97.7** W.h/day | |





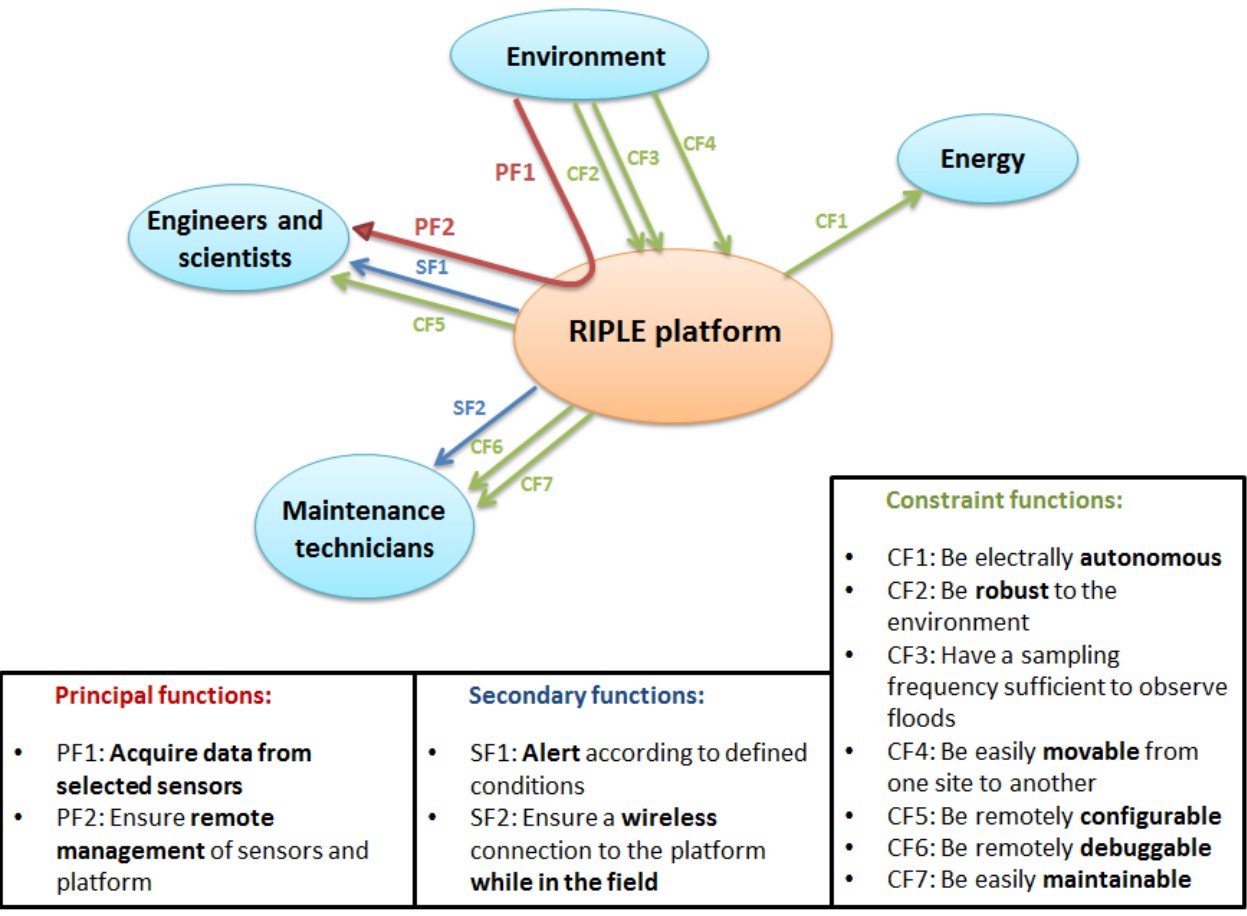

**Figure 1: Diagram of the functions to be taken into account for the design of the RIPLE platform.**







\* LSPIV : Large Scale Particle Image Velocimetry

\*\* SCAF : Suspended aggregates and flocs caracterisation system

**Figure 2: RIPLE platform architecture.**



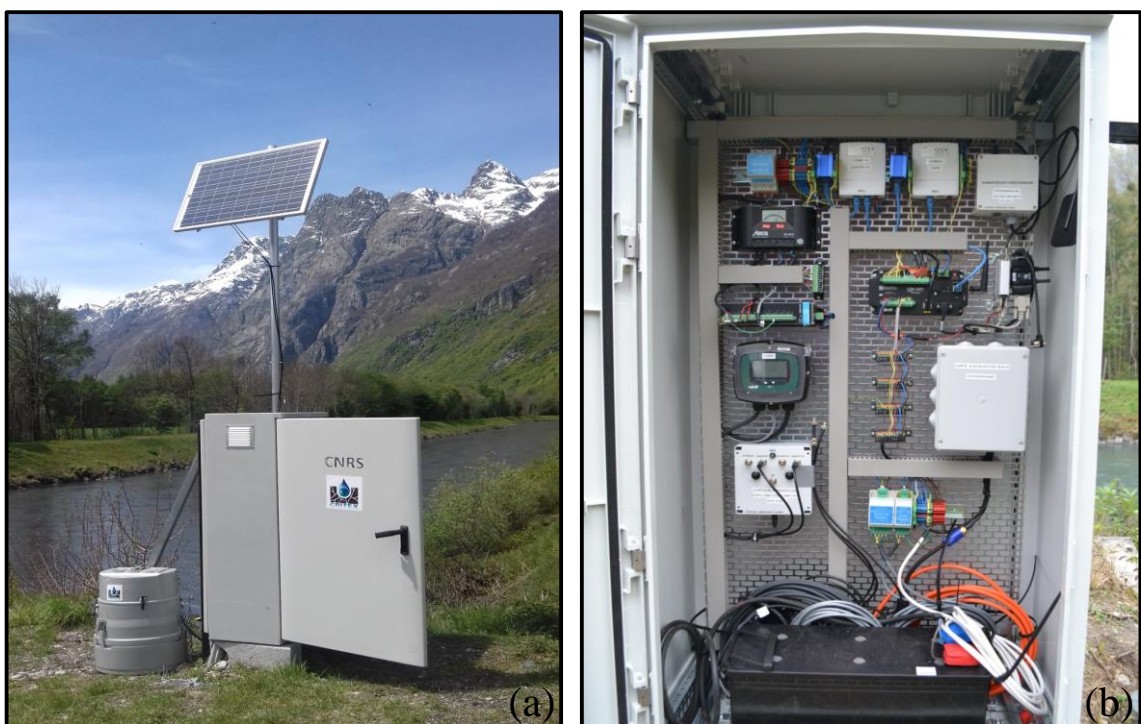

**Figure 3 : Photos of the box that houses all the electronics of the platform and the battery: (a) outside view where we also see the solar panel and the ISCO automatic river water sampler, (b) inside view.**


**Figure 4: Photos illustrating the deployment of instruments in situ (a) submerged instruments (conductivity probe, turbidimeters and hydrophone) housed in the polypropylene tubes anchored to the river bank, (b) water level and surface velocity radars supported by extensible brackets fixed to the bridge parapet, (c) control digital camera and LSPIV digital camera fixed at the top of a mast located on a river bank.**



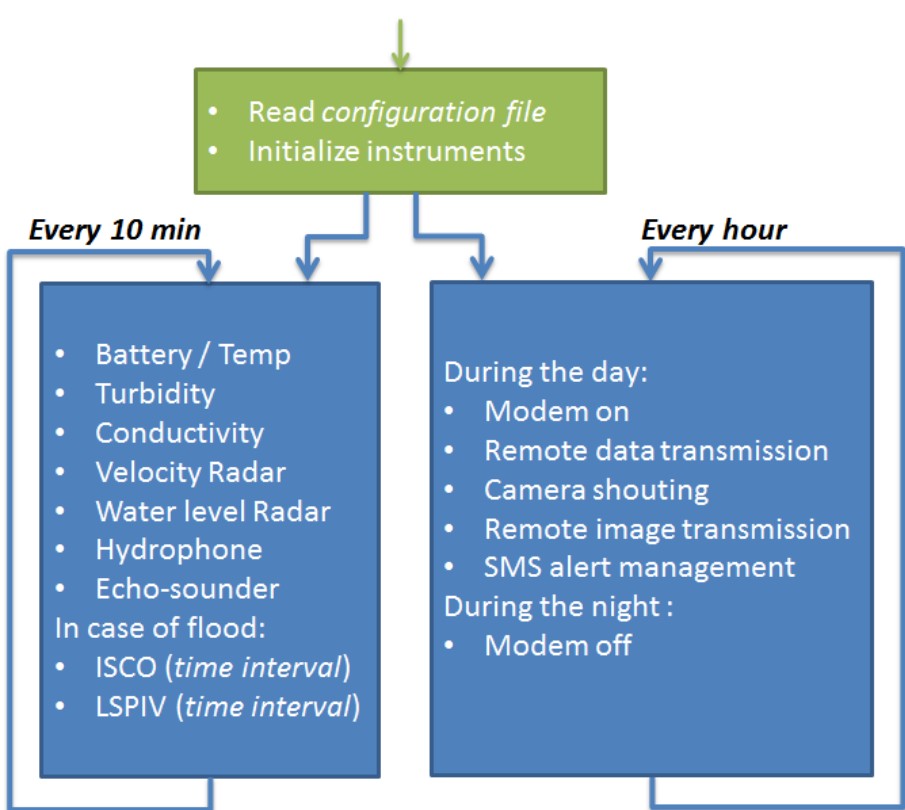

**Figure 5: Architecture of the main program that controls the RIPLE platform**





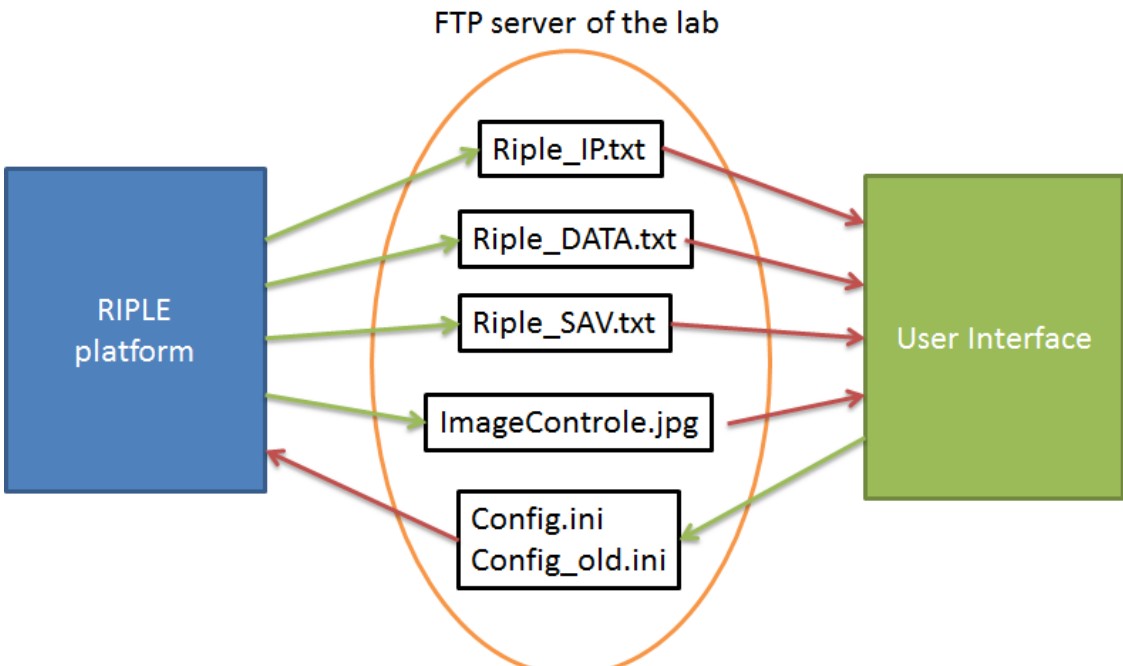

**Figure 6: Diagram describing the links between the RIPLE platform, the remote server and the user interface**

The "Riple_IP.txt" file contains the current IP address (public and dynamic) of the RIPLE platform.

5  The "Riple_DATA.txt" file contains the data from each instrument of the RIPLE platform.

The "Riple_SAV.txt" file contains the data that enable to remotely control the proper functioning of the RIPLE platform.

"imageControle_Date.jpg" is the image file that data logger puts them on the FTP. A copy of the image is saved locally on the

camera in order to make a reliable archiving of the images (in case of malfunction of the remote transmission for example).

"Config.ini" is the configuration file of the platform, which is generated when the user wants to change the configuration of

10  the platform from a remote location using the RIPLE interface (section 5). A backup of the old configuration is made in the

"Config_old.ini" file.





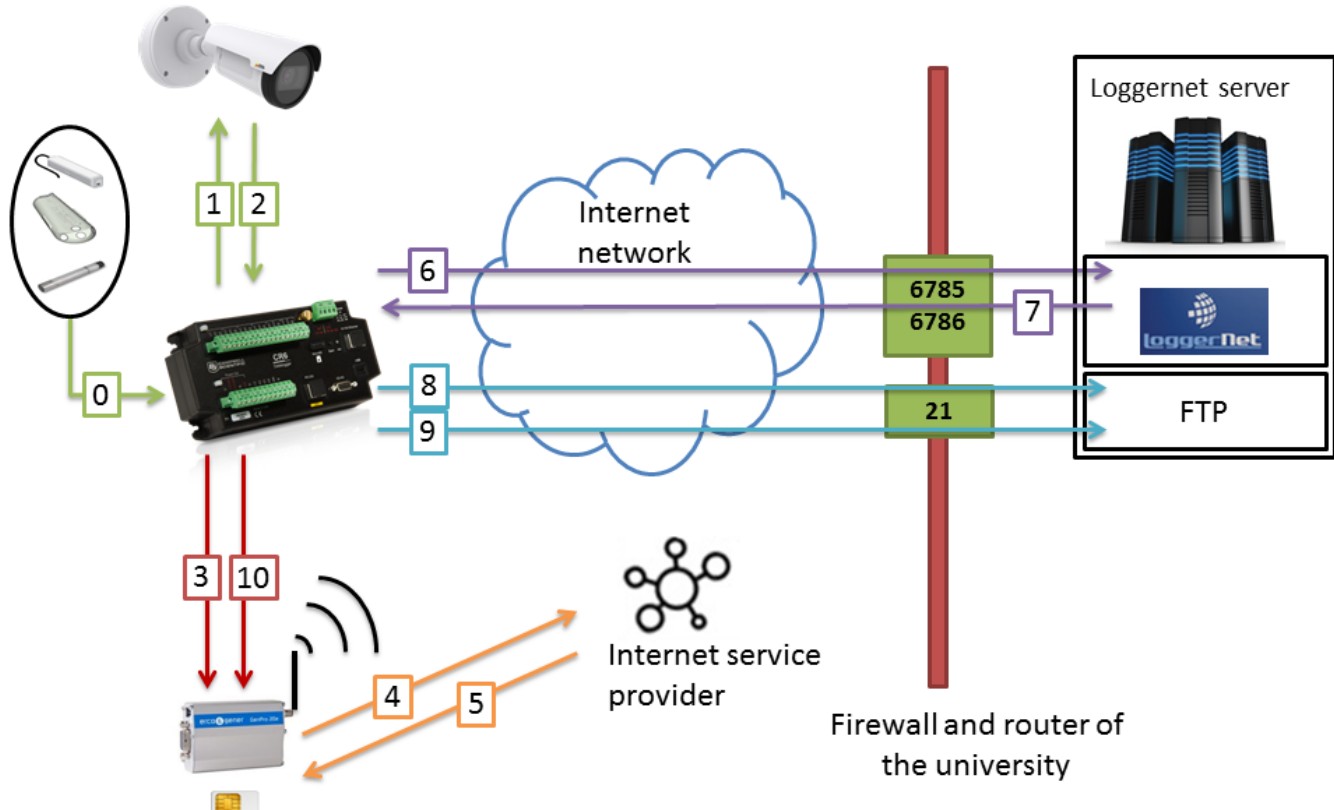

**Figure 7: Diagram of the data collection and data transfer to the remote FTP server**

0.     The data logger continuously retrieves data from the instruments integrated in RIPLE and stores the result in a table.

1.     The data logger supplies power to the control camera for a few minutes, which automatically takes an image when it starts up.

2.     The camera, configured as an FTP client, places the image on the data logger, which includes an FTP server.

3.     The data logger supplies power to the GPRS modem.

4.     The modem queries the Internet service provider to obtain a private IP address.

5.     The data logger is then connected to the 2.5G, 3G, 4G network.

6.     The data logger contacts the Loggernet server via port 6786.

7.     The Loggernet software running on the Loggernet server recognizes the data logger with its Packbus number, the connection is established. It is now possible for the manager of the platform to communicate remotely with the data logger.

8.     The data logger sends its data tables to the FTP server of the laboratory (FTPClient instruction).

9.     The data logger transmits the control image to the FTP server of the laboratory (FTPClient instruction).

10.     The data logger switches off the modem after a few hours.

11.     A web page created under Shiny (R package) is updated after each new data transmission.







**Figure 8: Screenshot of the "data visualization menu" of the user interface in the default mode. Four time series graphs are displayed from top to bottom: (1) water level and surface water velocity, (2) water level and turbidity, (3) water level and temperature, (4) water level and conductivity**



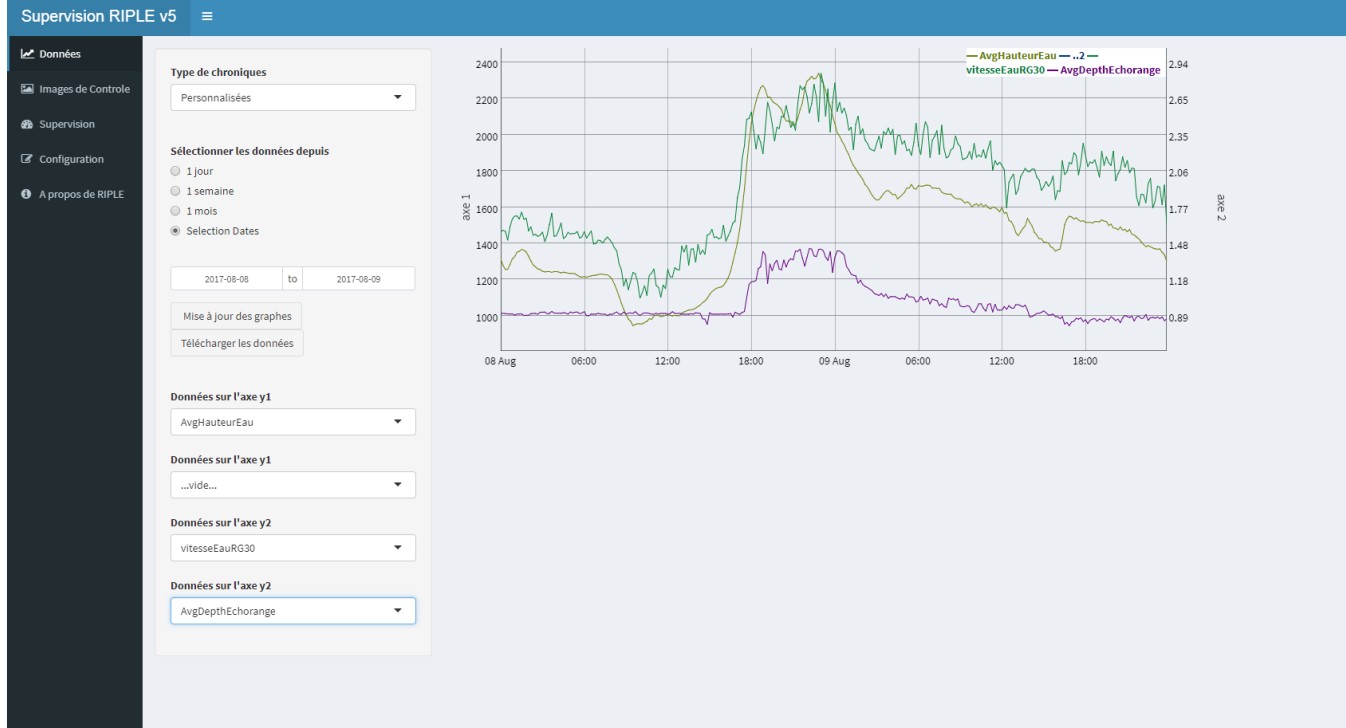

**Figure 9: Screenshot of the "data visualization menu" of the user interface in the personalized mode. A graph can be created with two variables on the 1st y-axis, two variables on the 2nd y-axis and in abscissa the time. In this case, the water level, water surface velocity and distance measured by the echo sounder are displayed.**



**Figure 10: Screenshots of the "images of control menu" of the user interface at two dates: (a) 06/11/2018 12:00 (UTC) for low water conditions, (b) 19/12/2018 10:00 (UTC) during a flood**





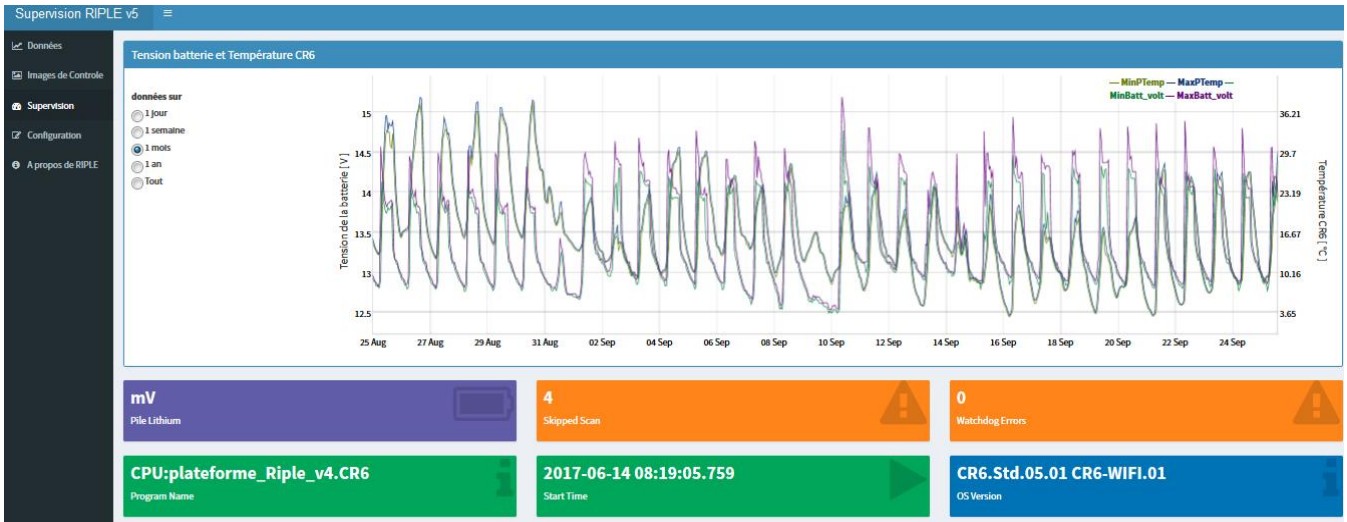

**Figure 11: Screenshots of the "supervision menu" of the user interface. A time series graph is displayed with minimum and maximum**
5   **battery voltage on the 1st y-axis, min and max air temperature on the 2nd y-axis**


**Figure 12: Screenshots of the "configuration menu" of the user interface. All the external variables of the program that can be modified are given below.**

"Numéros de tel pour alertes SMS": List of phone numbers that receive SMS alerts sent by RIPLE platform.

5 "SEUIL_ALERT_PTEMP": Threshold on the temperature of the data logger [in °C] above which an SMS alert is sent.

"SEUIL_ALERTE_BATTERIE": Threshold on the battery voltage [in V] below which an SMS alert is sent.

"SEUIL_VENTILO_PTEMP": Threshold on the temperature of the data logger [in °C] above which the fan is switched on.

"SEUIL_VENTILO_BATTERIE": Minimum battery voltage [in V] to allow the fan to work.

"HEURE_MODEM_ON": UTC time at which the RIPLE modem turns on each day, allowing data and images to be

10 transmitted remotely.

"HEURE_MODEM_OFF": UTC time at which the RIPLE modem turns off to limit power consumption.

"INTERVALLE_SMS": Time interval [in hours] between two SMS alerts.

"SEUIL_TURB_ACQ_LSPIV": Turbidity threshold [in FNU] above which the LSPIV digital camera takes a video.

"SEUIL_H_ACQ_LSPIV": Water level threshold in [mm] above which the LSPIV digital camera takes a video.





"INTERVALLE_LSPIV": Time interval between 2 consecutive LSPIV video acquisitions [in min]. Must be a multiple of the scan time (10 minutes in this study).

"HAUTEUR_CRUZOE": difference in elevation between the 0 of the staff gauge and the position of the radar [mm]. The water level is then calculated as the difference between "HAUTEUR_CRUZOE" and the distance measured by the radar.

5      "INTERVALLE_CLEAN_TURBI": Interval between two consecutive cleanings of the turbidimeter using a small brush [in hours].

"SEUIL_TURB_PRELEV_ISCO": Turbidity threshold above which the automatic water sampler starts its regular sampling [in FNU].

"SEUIL_H_PRELEV_ISCO": Water level threshold above which the automatic water sampler starts its regular sampling [in

10    mm].

"INTERVALLE_ISCO": Time interval between two samples [in min]. Must be a multiple of the scan time (10 minutes in this study).

"SEUIL_MEM_HDD": Threshold of the remaining memory on the hydrophone's HDD below which a collection must be planed [in GB].

The button « charger la configuration actuelle » triggers the reading of the "config.ini" file on the FTP by the data logger. This file contains the values of the variables currently loaded by RIPLE paltform.

The button « charger l'ancienne configuration » triggers the reading of the file "config_old.ini" on the FTP by the data logger, in which there is a backup of (n-1) configuration of RIPLE platform (in case of false manipulation).

20    The button « modifier la configuration » [ADMIN]: after entering the password, it allows to edit the value of each variable to change the RIPLE platform configuration.



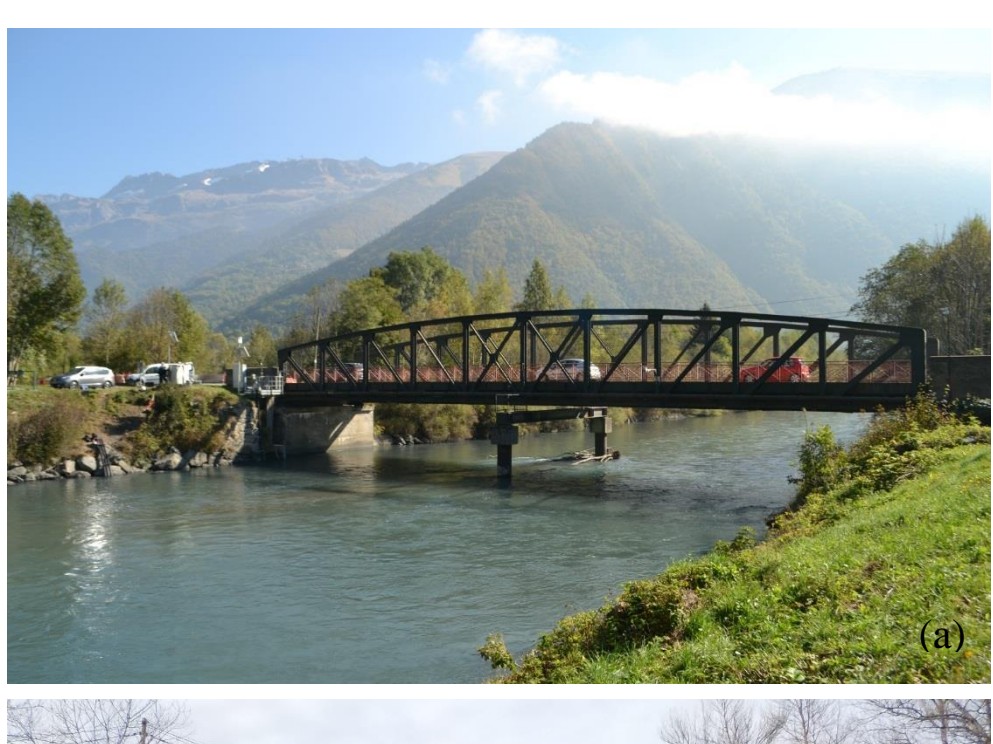

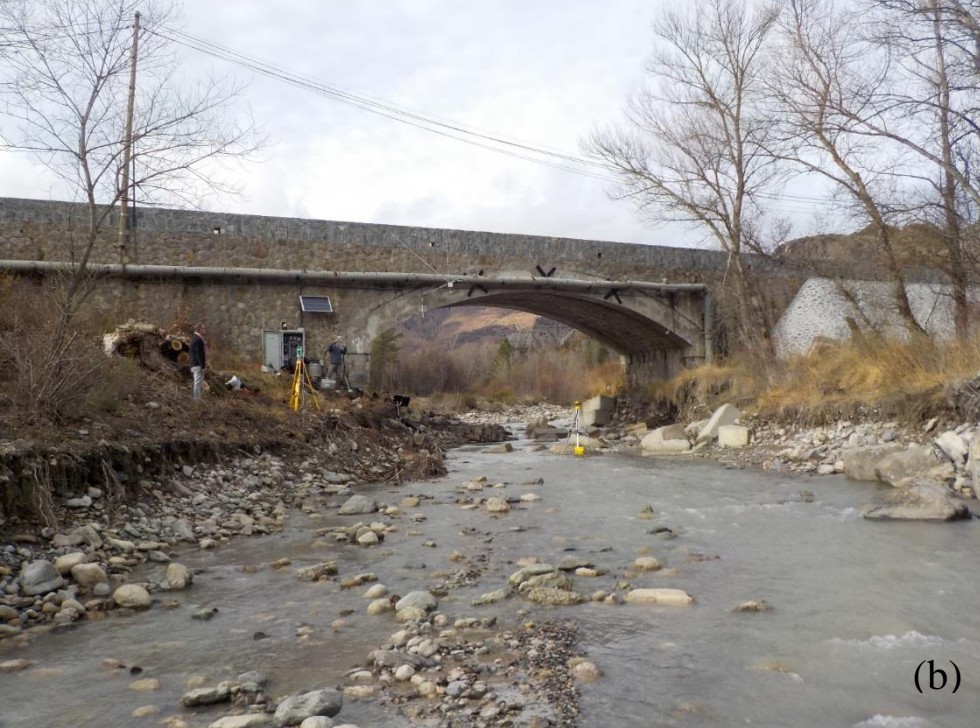

**Figure 13: Overview (from downstream) of the river section where the RIPLE platform is located on a) the Romanche in Bourg d'Oisans (45.1158°N, 6.0134°E, elevation 710 m) and b) the Galabre in La Robine sur Galabre (44.1586°N, 6.2360°E, elevation 680 m).**





## Appendix A: the electric diagram of RIPLE platform

