# Peer review of "An autonomous and low-power instrument platform for monitoring water and solid discharges in mesoscale rivers"

_Geoscientific Instrumentation, Methods and Data Systems, 2019_

## Referee Comment (RC1) · John Gray (Referee) · 5 Dec 2019

This is John R. Gray, peer reviewer of the subject manuscript, submitting the results of my review by:

1. Checking boxes on this website, and

2. Submitting a 7-page .pdf file that summarizes my NON-anonymous as a "supplement" on this website.

I debated selecting "accept as is," but figured there are minor issues that the authors might opt to address that could result in changes to the paper.

[Figure]

Please let me know if all of this information was not captured. Thanks for the opportunity to review this excellent paper. John

Please also note the supplement to this comment:
https://www.geosci-instrum-method-data-syst-discuss.net/gi-2019-33/gi-2019-33-RC1-supplement.pdf

**Supplement:**

**Date:** December 4, 2019

**From:** John R. Gray (U.S. Geological Survey Scientist Emeritus; Principal, GraySedimentology.com), Peer Reviewer

**To:** Guillaume Nord, Yoann Michielin, Romain Biron, Michel Esteves, Guilhem Freche, Thomas Geay, Alexandre Hauet, Cédric Legoût, and Bernard Mercier, Authors

**Through:** Natascha Töpfer, Copernicus Publications Editorial Support

**Subject: Peer Review of EGU Manuscript gi-2019-33, "An autonomous and low-power instrument platform for monitoring water and solid discharges in mesoscale rivers" (Geoscientific Instrumentation, Methods and Data Systems, European Geophysical Union)**

Thanks for the opportunity to review this manuscript on a subject Near and Dear to My Professional Heart. As the National Sedimentologist in the U.S. Geological Survey's (USGS) Office of Surface Water (1996-2014), my primary objective was to abet and augment the development of surrogate technologies for monitoring fluvial sediment and sorbed-constituent fluxes. In the 1990s, my then-USGS colleague Randy Parker suggested formulation of a "Sediment-Superga(u)ge Network" that would combine collection of hydrological data at a number of river sites, each using traditional and surrogate monitoring methods. A "Sediment Monitoring Instrumentation and Analysis Research Program (SMIARP)" among several U.S. Federal Agencies that encapsulated the Supergage concept ensued (Gray and Glysson, 2003; Gray, 2005). The RIPLE concept is a most-welcome and promising extension to the "Supergage" concept.

I have completed the on-line evaluation of the manuscript. As is my professional habit of 40+ years, I provide "Notable Observations and Considerations" before sharing my detailed review comments per page/line numbers. These are followed by my "references cited," which are provided solely for your information with neither an explicit nor implicit "requirement" for your use or even review.

Aside…the review format greatly facilitated my review. Kudos to the editor et al.

Do not hesitate to follow up directly with me with respect to my review comments at GraySedimentology@gmail.com.

Without further ado…

**NOTABLE OBSERVATIONS AND CONSIDERATIONS**

1. First and foremost, it is evident that this the RIPLE project was thoughtfully conceived and carefully developed. The authors are commended for their obvious in-depth knowledge of sedimentology, inventiveness, writing, and exposition skills.
2. The figures, particularly the photographs, are top-notch. The project sites are neat, and the beauty of the surrounding area(s) is stunning. I suggest all authors decline compensation for this effort due to the stunning work environment and ability to have so much fun on-the-job. Seriously?
3. One important concept of the Sediment Monitoring Instrument and Analysis Program (SMIARP, see later) is the key requirement for ground-truth data to be obtained concurrently with surrogate data. If this theme appears in the paper, I've missed it. The importance of ground-truth data can't be overstated for current calibration purposes, and for historical-to-present-to-future comparisons of datasets obtained using different technologies. There are examples of major 'breaks' in sediment transport in the USA that were subsequently determined to spurious artifacts of the data-collection process (instruments and/or methods).
4. I was pleased to see uncertainty estimates included with most of the parameters being measured.
5. Turbidity monitoring may be useful, but eventually one should consider including (an admittedly expensive but quite useful) subaqueous laser device for SSC AND particle-size distributions determined volumetrically. See the LISST-100X (or newer version) by Sequoia Scientific Inc., Bellevue, Washington, USA. I am not a fan of turbidity as a suspended-sediment concentration (SSC) surrogate even if I am a co-author of the 2009 USGS "turbidity guidelines."
6. For SSC values exceeding ~20 g/L, and particularly for hyperconcentrated streamflows, consider the ~simple and ~inexpensive densimetric technicque (Brown and others, 2015).
7. For bedload monitoring, it would be appropriate to state that passive hydroacoustics are relevant "only to gravel-bedload transport," if it can't be proven otherwise (wasn't so a decade ago…see Gray and others, 2010).,
8. Much of the "spec"-type information was tedious. Table 2 captures the "power balance," but I wonder if information on uncertainties, etc. might be included in yet another table?

**DETAILS, DETAILS**

1-10: Suggest that the acronym "RIPLE" instead be spelled out and followed by (RIPLE) in the abstract.

1-13: Terms such as "mesoscale" (somewhat defined on page 16-2 as having depths <5m), "fine" and "coarse" sediment (some consider the division at 0.062 mm, some at 2 mm), etc. would benefit from (more-or-less) precise definition.

1-16: Hydrophones…no mention of the bed type in which this technology is applicable. Unless times have changed, use of hydrophones (and geophones) is largely limited to gravel-bed rivers. If so, it would be appropriate to delimit the use of the hydrophones in the text.

1-18: I am a Big Fan of "robustness," even if some accuracy is lost. However, regarding accuracy, most of the RIPLE sub-technologies are qualified by uncertainty values. This is a practical and tractable approach, with which I concur.

2-15: A good reference on "persistent turbidity" is available in USGS SIR 2007-5178, "Major Turbidity Events in the North Santiam River Basin, Oregon, Water Years 1999–2004" (https://pubs.usgs.gov/sir/2007/5178/index.html).

2-31: Fluxes are "estimated" or "computed," no? I.e., "Estimates of suspended-sediment flux are usually obtained…"

3-1: "…method to obtain …SSC measurement…is not currently available." Advances in hydroacoustic determinations of sediment fluxes have gained considerable traction in the USA. I can provide several references upon request.

3-4/5: I have measured many turbulent rivers and have often found substantial cross-sectional and vertical gradients in SSC. A not-best example of this can be found in figures 3 and 4 in Gray and O'Halloran (2015) (I can provide more dramatic horizontal gradients, and many examples of vertical gradients if pressed).

3-10/11: Can reference American Society for Testing and Materials, 2000, Standard test methods for determining sediment concentration in water samples: D 3977-97, vol. 11.02, Water (II) 395-400. I have a copy of this standard.

3-15: Concur.

3-19: No hyphen is needed after "hydraulically" given that it is an adverb and can only refer to "based." Picky, aren't I…?!

4-5: Might considering starting sentence with, "Knowledge…essential for…"

4-8: "hysteretic" might be favored (more descriptive) over "dynamic."

4-14: "substantial" might be favored over "significant." I try to limit the use of the latter to statistics.

4-20++: Believe more paragraph breaks would benefit the paper here and elsewhere. I'd be glad to suggest where to insert paragraph breaks upon request, but I'm confident the authors can attend to this on their own.

5-11: "…but little or none for monitoring…"?

5-20: Suggest express units either as l/s or as $m^3/s$.

5-23: Suggest "…commercially available equipment…" instead of "…company equipment…"

5-25/6: aDcps, if considered non-intrusive, arguably qualify as "existing to monitor SSC."

5-30/1: Suggest, "All 'recording' instruments should be controlled…" but upon reflection a few days later, ignore this suggestion, given the impossibility of controlling non-recording equipment remotely…?!

6/1-4: See "Sediment Monitoring Instrument and Analysis Program (SMIARP)" entry under "notable comments."

6/4: Table 1, very informative/good.

6/18 (also 12/27): I was a peer reviewer for the PASS sampler paper. Very useful device; I recommended its acceptance after several back-and-forths with the principal author (Nic a very good scientist). However, the PASS technology as originally published doesn't provide information that would enable reliable flux estimates of suspended-sediment transport.

6/29: Appendix A is succinct and quite informative. I am impressed.

6/31: 10-minute data-measurement/recording interval: My experience indicates that this should be fine even if the rising phase of the hydrograph might be less than an hour in duration (?) for a "mesoscale river."

6/31-2: Data are plural…as you indicate in 15/8... Elsewhere the verb is correctly plural. Picky reviewer?!

7/11: I smiled at power consumption "estimate" of 7,758 mA.h/day. Are four significant figures for an estimate appropriate, even if it is mathematically (excruciatingly) correct…? But overall the power-consumption considerations and calculations are Far More Involved than I every performed at any of my gauges. I just tried to "overpower" them…! Well done.

7/24: Excellent table 1 and figure 2. Your efforts on figures make what otherwise would be a "dry" paper easier and more enjoyable to review.

7/27: At first, I was unsure whether this was an aerial or subaqueous camera. Easily discerned, but might be useful to start sentence with "The aerial control camera…" Suite yourselves.

8/30: Index velocity range is given, but value(s) used not so. Would it not be desirable to define "them" empirically with, say, a manually deployed aDcp? USGS has guidelines on the index-velocity method.

9/1: Wholeheartedly concur.

9/2: As I've aged (regrettably, I have considerable experience in that process), I've shied away from gender-specific references whenever possible. Instead of "manpower" I refer to "human resources." Yes, I know this is politically correct, but I'm the reviewer and that's what I wrote… 😊

9/6: Another foible, the USGS, for which I worked 37 years, mandates "relation" when used in scientific comparisons, and "relationship" to what I have with my wife, kids and grandson (born on October 26, 2019, I'm a rookie grandfather…). You of course aren't bound by USGS rules.

9/8: Reference is made to "…lack of brightness…" but I do not recall any reference to/caveats associated with "night monitoring." If this observation is correct and that some of the data could only be obtained half-the-time, it should be stated so. Perhaps in abstract and in body of text.

9/15: I found in the USGS "supergage" program that costs associated with human resources (!) required to operate and post-process surrogate data far outweighed capital costs of instruments. In fact, I called this my "Trojan Horse" project: I'd give instruments to eager-if-unsuspecting field folks who only later realized the extent of their commitment…! However, as time went on and the instruments were "figured out," such time commitments (and costs) diminished substantially.

10/15: Laser inclined upstream or downstream? And in a fixed position?

11/13: Radar, good.

11/14: Electrical conductivity versus Specific Conductivity: The latter is the "common" language, is indexed to a cm width of river water at 25 C (I recall).

12/21: Time averaged data for QW parameters is a good idea. Maybe not so for some physical parameters. I can explain if needed.

12/30: Suggest use, "including…" instead of ending sentence with "…"

13/8: Use of strainer good idea (a lesson I learned the hard way).

13/21++: Two turbidimeters at low and high range, very wise. However, I'm not a fan of turbidity to infer SSC even if I co-authored the "Rasmussen et al. (2009)" paper which we refer to as the USGS "turbidity protocol for estimating SSC." See "notable comment" on (expensive) alternative.

13/25: See "notable comments" on densimetric technique for high/hyperconcentrated streamflow SSC determinations.

15/3: Instead of "allows," perhaps "enables?"

15/5+: Suspended solids vs SSC: I avoid the former unless the metric is Total Suspended Solids (TSS), which I studied 2 decades ago and declared fundamentally unreliable. See Gray et al. 2000 and other papers that "condemn" TSS data.

15/8: SCAF is a very neat innovation. Congratulations. It in itself (with calibrating data) worth a paper?

15/27: Echo-sounder is at-a-point…?

17++ I neither reviewed  "data transfer" nor "visualization" closely, as I wouldn't have much to add given you're far ahead of me on these (and certainly other) topic(s).

22/7: "Sites" or "gauges" on two rivers?

23/11: My Ph.D. dissertation was on "calibration coefficients for 4 pressure-difference bedload samplers." I am keenly interested in using surrogate bedload data, but first things first (in USA, anyway) is to make sure we know trapping efficiencies of our ground-truth instruments. I found that some pressure-difference bedload samplers had substantially super-efficient trapping coefficients.

23/26: Dissolved Oxygen is (now) easily measured by luminescence technique, but of course its relevance is suspect-at-best in most non-tidal rivers.

24/6: Aquatic drone. Sounds neat, but how to use? I'm a rapt observer…

**References Cited:**

Brown, J.E., Gray, J.R., and Hornewer, N.J., 2015, In situ densimetric measurements as a surrogate for suspendedsediment concentrations in the Rio Puerco, New Mexico: Proceedings of the 3rd Joint Federal Interagency Conference (10th Federal Interagency Sedimentation Conference and 5th Federal Interagency Hydrologic Modeling Conference), April 19 – 23, 2015, Reno, Nevada, p. 1,261-1,272 (http://acwi.gov/sos/pubs/3rdJFIC/Contents/7BBrown.pdf and http://acwi.gov/sos/pubs/3rdJFIC/Proceedings.pdf).

Gray, J.R., and O'Halloran, Denis, 2015, Maximizing the reliability and cost- effectiveness of your suspendedsediment data: Proceedings of the 3rd Joint Federal Interagency Conference (10th Federal Interagency Sedimentation Conference and 5th Federal Interagency Hydrologic Modeling Conference), April 19 – 23, 2015, Reno, Nevada., p. 433-446 (http://acwi.gov/sos/pubs/3rdJFIC/Contents/3C-Gray.pdf and http://acwi.gov/sos/pubs/3rdJFIC/Proceedings.pdf).

Gray, J.R., (ed), 2005, Proceedings of the Federal Interagency Sediment Monitoring Instrument and Analysis Research Workshop, September 9-11, 2003, Flagstaff, Arizona: U.S. Geological Survey Circular 1276, 46 p. (http://water.usgs.gov/pubs/circ/2005/circ1276/).

Gray, J.R., and Glysson, G.D., 2005, Attributes for a sediment monitoring instrument and analysis research program, in, Proceedings of the Federal Interagency Sediment Monitoring Instrument and Analysis Workshop, September 9- 11, 2003, Flagstaff, Arizona, J.R. Gray, ed.: U.S. Geological Survey Circular 1276, 6 p. (http://water.usgs.gov/osw/techniques/sediment/sedsurrogate2003workshop/gray_glysson.pdf).

Gray, J.R., and Glysson, G.D., (eds), Proceedings of the Federal Interagency Workshop on Turbidity and Other Sediment Surrogates, April 30-May 2, 2002, Reno, Nevada, USA: U.S. Survey Circular 1250, 56 p. (https://pubs.usgs.gov/circ/2003/circ1250/).

Gray, J.R., Glysson, G.D., Turcios, L.M., and Schwarz, G.E., 2000, Comparability of suspended-sediment concentration and total suspended solids data: U.S. Geological Survey Water-Resources Investigations Report 00- 4191, 14 p. (http://water.usgs.gov/osw/pubs/WRIR00-4191.pdf).

**Uncited References:**

See GraySedimentology.com bio/bibliography for other papers of potential interest…the following in particular (if the link is broken, I can forward the paper).

Gray J.R., and Landers M.N., 2014, Measuring suspended sediment. In: Ahuja S. (ed.) Comprehensive Water Quality and Purification: United States of America, Elsevier, vol. 1, p. 157-204 (http://water.usgs.gov/osw/techniques/sediment/gray_landers_elsevier_chapter_12_10_17_2013.pdf).

---

## Author Comment (AC1) · 20 Dec 2019

**Author comments to referee 1 (John R. Gray) comments on "An autonomous and low-power instrument platform for monitoring water and solid discharges in mesoscale rivers" by Nord et al.**

In the following, the reviewer comments appear in black italic and our answers are provided in blue. When there are quotations from the text of the article, they appear in quotation marks and the new or corrected parts are highlighted in yellow.

gi-2019-33-RC1.pdf

*This is John R. Gray, peer reviewer of the subject manuscript, submitting the results of my review by:*

*1. Checking boxes on this website, and*

*2. Submitting a 7-page .pdf file that summarizes my NON-anonymous as a "supplement" on this website.*

*I debated selecting "accept as is," but figured there are minor issues that the authors might opt to address that could result in changes to the paper.*

*Please let me know if all of this information was not captured. Thanks for the opportunity to review this excellent paper. John*

*Please also note the supplement to this comment:*

*https://www.geosci-instrum-method-data-syst-discuss.net/gi-2019-33/gi-2019-33-RC1-supplement.pdf*

Answer: we sincerely thank John R. Gray for his very careful review of this paper. We acknowledge his extensive experience in measuring liquid and solid fluxes (fine and coarse sediments) in rivers. We have tried to take his considerations into account as much as possible. We have replied below to his comments

gi-2019-33-RC1-supplement.pdf

*Thanks for the opportunity to review this manuscript on a subject Near and Dear to My Professional Heart. As the National Sedimentologist in the U.S. Geological Survey's (USGS) Office of Surface Water (1996-2014), my primary objective was to abet and augment the development of surrogate technologies for monitoring fluvial sediment and sorbed-constituent fluxes. In the 1990s, my then-USGS colleague Randy Parker suggested formulation of a "Sediment-Superga(u)ge Network" that would combine collection of hydrological data at a number of river sites, each using traditional and surrogate monitoring methods. A "Sediment Monitoring Instrumentation and Analysis Research Program (SMIARP)" among several U.S. Federal Agencies that encapsulated the Supergage concept ensued (Gray and Glysson, 2003; Gray, 2005). The RIPLE concept is a most-welcome and promising extension to the "Supergage" concept.*

*I have completed the on-line evaluation of the manuscript. As is my professional habit of 40+ years, I provide "Notable Observations and Considerations" before sharing my detailed review comments per page/line numbers. These are followed by my "references cited," which are provided solely for your information with neither an explicit nor implicit "requirement" for your use or even review.*

*Aside…the review format greatly facilitated my review. Kudos to the editor et al.*

*Do not hesitate to follow up directly with me with respect to my review comments at GraySedimentology@gmail.com.*

*Without further ado…*

Answer: once again, we thank John R. Gray for his professionalism and insightful comments. We also thank him for the bibliographical references that include the direct links to the related documents. We respond to the Notable Observations and Considerations in the next section and to the detailed review comments thereafter.

NOTABLE OBSERVATIONS AND CONSIDERATIONS

*1. First and foremost, it is evident that this the RIPLE project was thoughtfully conceived and carefully developed. The authors are commended for their obvious in-depth knowledge of sedimentology, inventiveness, writing, and exposition skills.*

Answer: we are pleased that the development of this platform meets with the current quality standards of the scientific community and more specifically in the field of sedimentology.

As cited in the text, the development of this platform was possible thanks to a fifteen-year expertise in hydrometric and sediment measurements within the IGE laboratory and more broadly within the research laboratories in the Grenoble and Lyon communities and local companies such as EDF. This work was supported by the EQUIPEX CRITEX project (grant no. ANR-11-EQPX-0011, PIs J. Gaillardet and L. Longuevergne) whose main objective was to build a national park of instruments for the study of the Critical Zone (https://www.critex.fr/). Yoann Michielin benefited from an engineer contract (1.5 years) from the CRITEX project to develop the RIPLE platform.

*2. The figures, particularly the photographs, are top-notch. The project sites are neat, and the beauty of the surrounding area(s) is stunning. I suggest all authors decline compensation for this effort due to the stunning work environment and ability to have so much fun on-the-job. Seriously?*

Answer: It is true that we work in a very pleasant environment. The measurement sites we usually follow are located in south-eastern France (Nord et al., 2017; Esteves et al., 2019), mountain regions subject to Mediterranean influences where erosion is significant. This aesthetic environment makes it possible to compensate for the efforts and sometimes the ingratitude involved in monitoring these morphologically active and very dynamic rivers.

*3. One important concept of the Sediment Monitoring Instrument and Analysis Program (SMIARP, see later) is the key requirement for ground-truth data to be obtained concurrently with surrogate data. If this theme appears in the paper, I've missed it. The importance of ground-truth data can't be overstated for current calibration purposes, and for historical-to-present-to-future comparisons of*

*datasets obtained using different technologies. There are examples of major 'breaks' in sediment transport in the USA that were subsequently determined to spurious artifacts of the data-collection process (instruments and/or methods).*

Answer: For suspended sediment monitoring, we systematically take water samples during floods to measure directly the suspended sediment concentration (SSC) and establish turbidity-SSC relations. Therefore, there are ground-truth data. This is fundamental in these mesoscale rivers where the nature of the transported materials varies greatly depending on the sources of erosion activated within the watershed. Indeed turbidity-SSC relations are particularly sensitive to the size of the materials transported and can change from one flood to another or even during a flood. In addition, the characterization of the sediment settling velocity using SCAF (sediment settling velocity characterization device) is carried out in situ, directly after sampling, to avoid flocculation/disaggregation problems that may occur later and the inconvenience of sub-sampling.

We believe that the comment of the referee refers to flow discharge measurements (gauging) on the one hand and direct measurements of bed load fluxes on the other. Concerning flow discharge measurements, our publications (e.g. Nord et al., 2017; Esteves et al., 2019) show that we have spent a lot of time and energy carrying out gauging, including during floods. This involves measurements by portable SVR, LSPIV, dilution method, current meter taking into account that the sites are remote from the lab (2 or 3 hour drive) and subject to with very short hydrological reactions. We continue these measurements regularly at our sites. However, we recognize that it is almost impossible to maintain traditional stage-discharge rating curves in very morphologically active rivers. This is why we have been interested in developing almost direct flow discharge estimation methods in recent years (Tafasca, 2017; Hasanyar, 2019).

With regard to bed load fluxes measurements, we recognize that they have not been performed yet for the sites presented in this study. We are considering carrying out some measurement campaigns by mobile bag samplers but the success is very uncertain on this very turbulent and dynamic type of river. Direct monitoring of bed load fluxes is not included in the RIPLE platform since there is no method that can be applied in mesoscale rivers. There have indeed been attempts to monitor bedload fluxes by direct methods in our immediate environment, but limited to small rivers (Liebault et al., 2016). Within the Draix observatory, a monitoring of total event bed load yield is carried out (Mathys, 2006) by emptying sediment traps, but once again, this is not possible for mesoscale rivers. The passive hydro-acoustic method integrated into the RIPLE platform has been verified against alternative methods or ground-truth data at other sites (Geay et al., 2017a; Geay et al., 2017b). More recently, we have complemented the monitoring by passive seismic at RIPLE sites. To date, our objective is not to produce an estimation of bedload fluxes but rather to be able to say whether or not there is bedload transport and possibly to characterize a level of amplitude of the phenomenon.

*4. I was pleased to see uncertainty estimates included with most of the parameters being measured.*

Answer: we agree with this comment.

*5. Turbidity monitoring may be useful, but eventually one should consider including (an admittedly expensive but quite useful) subaqueous laser device for SSC AND particle-size distributions determined volumetrically. See the LISST-100X (or newer version) by Sequoia Scientific Inc., Bellevue, Washington,*

*USA. I am not a fan of turbidity as a suspended-sediment concentration (SSC) surrogate even if I am a co-author of the 2009 USGS "turbidity guidelines."*

Answer: We did consider this technology. However, the LIST is not suitable for rivers with SSC values typically higher than a few g/l. Moreover, the SCAF device was developed to meet metrological needs in rivers with high SSC values. We know that there has been an attempt to develop a sub-sampling system to carry out in situ monitoring with a LIST in a tropical river with high SSC values during floods (Lajeunesse et al., 2018). However, such a system has not been industrialized and is relatively heavy in maintenance. As soon as there is a validated and available system, we will be ready to integrate it into the RIPLE platform.

*6. For SSC values exceeding ~20 g/L, and particularly for hyperconcentrated streamflows, consider the ~simple and ~inexpensive densimetric technicque (Brown and others, 2015).*

Answer: we thank John R. Gray for this information and the corresponding reference. It is a technology we should test in the future. The only limitation we see is that it does not allow low SSC to be measured. This means that several measurement systems are required to cover the entire SSC range and then combine the different measurements.

In the site where RIPLE is currently installed (Galabre), SSC values can reach typically several tens of g/l. The commercial turbidimeters we have selected, which operate in backscattering (MJK Susix or previously WTW Visolid 700 IQ), are designed to operate in high SSC ranges. Our experience is more limited for the MJK Susix model but the WTW Visolid 700 IQ model has given good results for SSC of more than 100 g/l (see fig.3 Navratil et al., 2011). This allows us to have continuous timeseries of turbidity using a single instrument for both low and high SSC values. We recognise that it is still necessary to take automatic samples to calibrate the turbidity-SSC relations.

It should also be noted that the so-called "capteur marseillais" turbidimeter, integrated into the RIPLE platform, was originally designed for very high SSC ranges. It has been operating in Draix since 1994 and has measured turbidity for SSC above 500 g/l (Bergougnoux et al., 1998).

*7. For bedload monitoring, it would be appropriate to state that passive hydroacoustics are relevant "only to gravel-bedload transport," if it can't be proven otherwise (wasn't so a decade ago…see Gray and others, 2010).,*

Answer: Indeed, hydrophones have been mostly tested in gravel-bed rivers. This statement has been precised (3/30) as follows: "The acoustic power of bedload sounds has been related to bedload fluxes by using site-specific calibration curves in gravel-bed rivers". However, we shall mention that Thorne used hydrophones for monitoring the bedload transport of sand in marine environment [Diameters>0.1 mm]. See for example a review of his work (Thorne, 2014). If this is theoretically possible to record the sound generated by sand impacts, it is however limited by the type of river in which the measurements are done. Indeed, sand impacts generates high frequency sounds. This high frequency sounds do not propagate in all rivers, high frequency sounds are heavily attenuated in steep rivers (see Geay et al., 2019). However, in sand rivers or marine environment, the sound of sand particles can be used to monitor bedload transport. Finally, we can not state that the use of hydrophones is "largely limited" to gravel-bed rivers.

*8. Much of the "spec"-type information was tedious. Table 2 captures the "power balance," but I wonder if information on uncertainties, etc. might be included in yet another table?*

Answer: we could add a Table with the uncertainties of the sensors already mentioned in the text but we are not convinced of the added-value of this Table. Indeed, the actual uncertainty, in the case of a sensor installed in the field, is often much higher than the uncertainty value given by the manufacturer which corresponds to an idealized situation. In the field, there are many other sources of uncertainty that should be taken into account and this requires a specific work.

*DETAILS, DETAILS*

*1-10: Suggest that the acronym "RIPLE" instead be spelled out and followed by (RIPLE) in the abstract.*

Answer: Yes thanks, it's a mistake on our part. This has been corrected in the abstract.

*1-13: Terms such as "mesoscale" (somewhat defined on page 16-2 as having depths <5m), "fine" and "coarse" sediment (some consider the division at 0.062 mm, some at 2 mm), etc. would benefit from (more-or-less) precise definition.*

Answer: By mesoscale rivers, we refer to rivers which drain mesoscale catchments, i.e. from 10 to 1000 km² (Singh et al., 2018). It is true that this information was not specified in the text. This has been added at the end of the first sentence of the abstract.

In this study, the boundary between fine and coarse sediments, which also applies to distinguish between suspended and bedload transport, refers to sand-sized particles. This limit, arbitrarily fixed around 2 mm, is explained by the fact that the suction velocity of the automatic sampler is 0.58 m/s for a suction height of 4.6 m (data provided by ISCO for the 3700 model), which does not allow to collect particles coarser than a few millimeters. We sometimes find sand-sized particles in river samples, but they are rare. Representative sampling of sand-sized particles remains a very topical and unresolved issue to date. We have added this sentence in the introduction: "In this study, suspension load applies for particles finer than 2 mm."

*1-16: Hydrophones…no mention of the bed type in which this technology is applicable. Unless times have changed, use of hydrophones (and geophones) is largely limited to gravel-bed rivers. If so, it would be appropriate to delimit the use of the hydrophones in the text.*

Answer: See our response to comment 7 in "notable observations and considerations".

*1-18: I am a Big Fan of "robustness," even if some accuracy is lost. However, regarding accuracy, most of the RIPLE sub-technologies are qualified by uncertainty values. This is a practical and tractable approach, with which I concur.*

Answer: we agree with this comment.

*2-15: A good reference on "persistent turbidity" is available in USGS SIR 2007-5178, "Major Turbidity Events in the North Santiam River Basin, Oregon, Water Years 1999–2004" (https://pubs.usgs.gov/sir/2007/5178/index.html).*

Answer: we agree with this comment. This reference has been added to the text.

*2-31: Fluxes are "estimated" or "computed," no? I.e., "Estimates of suspended-sediment flux are usually obtained…"*

Answer: ok, the sentence has been corrected as followed: "Estimates of suspended sediment flux are usually obtained by multiplying the water discharge by the SSC…"

*3-1: "…method to obtain …SSC measurement…is not currently available." Advances in hydroacoustic determinations of sediment fluxes have gained considerable traction in the USA. I can provide several references upon request.*

Answer: We are aware that hydroacoustic is one of the tools that will make possible in the coming years to estimate sediment fluxes by class of particle size thanks to the use of multi-frequency technology. In fact, we had considered integrating the Aquascat instrument (by Aquatec) into the RIPLE platform (see Table 2). However, at present, to the best of our knowledge (Thorne and Hurther, 2014; Vergne et al., 2017), it is still necessary to develop theoretical bases to inverse the acoustic signal under natural flow conditions with a wide range of particle sizes of non-spherical shape and in the presence of air bubbles. As soon as the theoretical framework allows this type of instrument to be used routinely in rivers, we will integrate hydroacoustic into RIPLE for sediment fluxes estimation. Nevertheless, we are interested in the references that can be provided by the referee.

*3-4/5: I have measured many turbulent rivers and have often found substantial cross-sectional and vertical gradients in SSC. A not-best example of this can be found in figures 3 and 4 in Gray and O'Halloran (2015) (I can provide more dramatic horizontal gradients, and many examples of vertical gradients if pressed).*

Answer: We recognize that the spatial distributions of SSC across the cross section of the river may be heterogeneous in some cases. However, we hypothesize that in mountainous mesoscale rivers where the bed slope is greater than 0.1% and where the turbulence generated during floods is sufficiently high, the spatial distribution of SSC is relatively homogeneous. This hypothesis was partially validated by Mano (2008) during his PhD on the Isère river in Grenoble using manual samples during a hydraulic dam flushing and for SSC values of a few g/l. It is true that this question of the spatial distribution of SSC across the section remains a scientific issue, particularly in rivers where the SSC is greater than a few g/l. This issue will remain a barrier until we have the means to spatially sample suspended sediment fluxes. Hydroacoustic will be able to provide some keys to these locks but probably not in the case of high SSC values (due to potential attenuation) and in rivers subjected to flash floods with damaging impact.

*3-10/11: Can reference American Society for Testing and Materials, 2000, Standard test methods for determining sediment concentration in water samples: D 3977-97, vol. 11.02, Water (II) 395-400. I have a copy of this standard.*

Answer: we agree with this comment. We added the following reference in the text: "International Organization for Standardization, 2005: Liquid flow measurement in open channels – Sediment in streams and canals — Determination of concentration, particle size distribution and relative density. ISO 4365, Geneva."

*3-15: Concur.*

Answer: no comment.

*3-19: No hyphen is needed after "hydraulically" given that it is an adverb and can only refer to "based." Picky, aren't I…?!*

Answer: Yes, thanks for that precision. The correction has been made in the text.

*4-5: Might considering starting sentence with, "Knowledge…essential for…"*

Answer: OK, thanks for the proposition. The sentence was corrected as follows: "Knowledge of water discharge is essential whether it is for estimating suspended sediment fluxes, dissolved matter fluxes, nutrient or contaminant fluxes associated with fine particles."

*4-8: "hysteretic" might be favored (more descriptive) over "dynamic."*

Answer: OK, thanks for the proposition. The sentence has been corrected as follows: "… solid discharges can demonstrate a hysteretic and transitory behavior, more impulsive than the flow itself, …"

*4-14: "substantial" might be favored over "significant." I try to limit the use of the latter to statistics.*

Answer: "significant" was replaced by "substantial".

*4-20++: Believe more paragraph breaks would benefit the paper here and elsewhere. I'd be glad to suggest where to insert paragraph breaks upon request, but I'm confident the authors can attend to this on their own.*

Answer: OK, a paragraph break has been added. A new paragraph starts with: "The first variable that has been added to hydrometric stations is velocity."

Elsewhere in the text, there are paragraph breaks that may not be visible because there is no line break. This has been fixed.

*5-11: "…but little or none for monitoring…"?*

Answer: Thanks, this has been corrected in the text.

*5-20: Suggest express units either as l/s or as m 3 /s.*

Answer: we agree with this comment. "(from a few dozen l/s to several hundred of $m^3/s$)" has been replaced by "(from $10^{-2}$ to $10^2$ $m^3/s$)".

*5-23: Suggest "…commercially available equipment…" instead of "…company equipment…"*

Answer: thanks. "company equipment" has been replaced by "commercially available equipment".

*5-25/6: aDcps, if considered non-intrusive, arguably qualify as "existing to monitor SSC."*

Answer: We do not consider aDcp as a non-intrusive device available to monitor SSC. Furthermore, we justify the choice not to use aDcp for different reasons:

- moving-boat aDcp is difficult to consider for monitoring. Moreover application of moving-boat aDcps to steep rivers (slope >1%) during flood is not feasible due to waves on the water surface that prevent the permanent contact of the aDcp with the water surface and due to floating debris.

- alternatively V-aDcps such as the Sontek IQ Plus (see Table 2) can be integrated in the RIPLE platform. This was done in the Claduègne river by Nord et al. (2017). Such device, which is fixed to the bottom of the river, can be used to monitor velocity profile in a specific vertical of the cross section. However it is an intrusive instrument and its installation is hazardous depending on the type of river. Furthermore, this model does not provide the backscattered intensity, which is necessary to derive SSC.

- signal attenuation may be strong when SSC values are high (typically higher than a few g/l), which may prevent to estimate velocity and SSC (Nord et al., 2014).

*5-30/1: Suggest, "All 'recording' instruments should be controlled…" but upon reflection a few days later, ignore this suggestion, given the impossibility of controlling non-recording equipment remotely…?!*

Answer: no comment.

*6/1-4: See "Sediment Monitoring Instrument and Analysis Program (SMIARP)" entry under "notable comments."*

Answer: See our response to comment 3 in "notable observations and considerations".

*6/4: Table 1, very informative/good.*

Answer: OK, thanks.

*6/18 (also 12/27): I was a peer reviewer for the PASS sampler paper. Very useful device; I recommended its acceptance after several back-and-forths with the principal author (Nic a very good scientist). However, the PASS technology as originally published doesn't provide information that would enable reliable flux estimates of suspended-sediment transport.*

Answer: we agree with this comment.

*6/29: Appendix A is succinct and quite informative. I am impressed.*

Answer: OK, thanks.

*6/31: 10-minute data-measurement/recording interval: My experience indicates that this should be fine even if the rising phase of the hydrograph might be less than an hour in duration (?) for a "mesoscale river."*

Answer: we agree with this comment. The rising phase can be very fast (typically one hour) in steep rivers that give rise to flash floods. The 10-minute time step is a good compromise between representative sampling and the time required to scan all instruments integrated in RIPLE.

*6/31-2: Data are plural…as you indicate in 15/8... Elsewhere the verb is correctly plural. Picky reviewer?!*

Answer: OK, thanks for your correction. The text has been changed accordingly.

*7/11: I smiled at power consumption "estimate" of 7,758 mA.h/day. Are four significant figures for an estimate appropriate, even if it is mathematically (excruciatingly) correct…? But overall the power-consumption considerations and calculations are Far More Involved than I every performed at any of my gauges. I just tried to "overpower" them…! Well done.*

Answer: we agree with this comment but this does not seem problematic to us since the power consumptions are expressed in mA.h/day.

*7/24: Excellent table 1 and figure 2. Your efforts on figures make what otherwise would be a "dry" paper easier and more enjoyable to review.*

Answer: OK, thanks.

*7/27: At first, I was unsure whether this was an aerial or subaqueous camera. Easily discerned, but might be useful to start sentence with "The aerial control camera…" Suite yourselves.*

Answer: OK, the word "aerial" was added to the sentence.

*8/30: Index velocity range is given, but value(s) used not so. Would it not be desirable to define "them" empirically with, say, a manually deployed aDcp? USGS has guidelines on the index-velocity method.*

Answer: we understand the comment of the reviewer. The procedure indicated in the text is very generic. The values of the coefficient that links the average velocity in a vertical to the corresponding surface velocity are taken from the literature. We are aware that it is preferable to apply the USGS guidelines. This was actually done in our cases of application (Nord et al., 2014; Tafasca, 2017) where aDcp or current meter gaugings were available.

The sentence was complemented as follows: "A transect of surface velocity along the cross section of the river is extracted and converted to a transect of depth averaged velocity over the vertical using a coefficient that relates the depth averaged velocity to the surface velocity. Such coefficient commonly ranges between 0.75 and 0.85 (Hauet et al., 2008; Le Coz et al., 2010; Welber et al., 2016) but it is preferable to define it from aDcp or current meter measurements."

*9/1: Wholeheartedly concur.*

Answer: OK, no comment.

*9/2: As I've aged (regrettably, I have considerable experience in that process), I've shied away from gender-specific references whenever possible. Instead of "manpower" I refer to "human resources." Yes, I know this is politically correct, but I'm the reviewer and that's what I wrote…*

Answer: Yes, that is a very good point. "Manpower" has been replaced by "human resources" in the text.

*9/6: Another foible, the USGS, for which I worked 37 years, mandates "relation" when used in scientific comparisons, and "relationship" to what I have with my wife, kids and grandson (born on October 26, 2019, I'm a rookie grandfather…). You of course aren't bound by USGS rules.*

Answer: it is noted. "Relationship" has been replaced by "relation".

*9/8: Reference is made to "…lack of brightness…" but I do not recall any reference to/caveats associated with "night monitoring." If this observation is correct and that some of the data could only be obtained half-the-time, it should be stated so. Perhaps in abstract and in body of text.*

Answer: The control camera is only activated during day time. We have completed the following sentence: "The triggering of the acquisition is programmed as follows: (i) the camera is started by the data logger only during day time via the electrical relay (typically once an hour)"

On the other hand, The LSPIV camera, is activated when triggering conditions of turbidity and water levels are overcome. For the sequences acquired during the night, an IR projector housed into the camera is used. However, the effectiveness of this projector depends on the distance between the camera and the river section. It is specified on page/line 10/9 that "primarily tests have shown that the projector is not powerful enough to illuminate up to the water surface." Future camera models or the use of independent IR projectors may improve this point. In the meantime, sequences

acquired during the night or in low light conditions are not used for LSPIV analysis. That is why the term "lack of brightness" was mentioned on page/line 9/8 for the manual selection of the video sequences to be used for LSPIV analyses.

*9/15: I found in the USGS "supergage" program that costs associated with human resources (!) required to operate and post-process surrogate data far outweighed capital costs of instruments. In fact, I called this my "Trojan Horse" project: I'd give instruments to eager-if-unsuspecting field folks who only later realized the extent of their commitment…! However, as time went on and the instruments were "figured out," such time commitments (and costs) diminished substantially.*

Answer: We fully agree with this comment. The monitoring of hydrometric variables is very time-consuming. Despite the automation of measurements, it is still necessary to visit the monitoring sites regularly, for example to check the state of the sensors, clean the sensors, adapt their position to the morphological changes in the bed…

*10/15: Laser inclined upstream or downstream? And in a fixed position?*

Answer: We are not very sure of the object to which this comment refers because the line indicated does not seem appropriate but we think that the reviewer refers to the surface velocity radar. The manufacturer recommends an orientation of the radar towards upstream, but this is not always optimal depending on the study sites. We have several times considered that the hydraulic conditions were more favorable by looking at downstream and this did not affect the quality of the measurements. On the other hand, the radar must be fixed and its position must remain unchanged over time.

*11/13: Radar, good.*

Answer: OK, no comment.

*11/14: Electrical conductivity versus Specific Conductivity: The latter is the "common" language, is indexed to a cm width of river water at 25 C (I recall).*

Answer: We are not specialists in this measurement. The information provided in the article comes from the technical documentation of the selected instrument. We prefer to leave the text as it is.

*12/21: Time averaged data for QW parameters is a good idea. Maybe not so for some physical parameters. I can explain if needed.*

Answer: We average the physical and chemical parameters over a short period of time, typically 30 s or less. A large number of measurements are made during this period at frequency ≥ 1 Hz and these measurements are finally averaged. This method avoids to be affected by turbulent fluctuations.

*12/30: Suggest use, "including…" instead of ending sentence with "…"*

Answer: OK thanks, this has been corrected in the text as follows: "… for all subsequent analyses of dissolved and particulate phases including major ions, nutrients, contaminants, microorganisms, DNA."

*13/8: Use of strainer good idea (a lesson I learned the hard way).*

Answer: Yes, this is recommended by the manufacturer and it is true that it prevents clogging the pipe that brings water to the pump of the sampler.

13/21++: Two turbidimeters at low and high range, very wise. However, I'm not a fan of turbidity to infer SSC even if I co-authored the "Rasmussen et al. (2009)" paper which we refer to as the USGS "turbidity protocol for estimating SSC." See "notable comment" on (expensive) alternative.

Answer: We explained the principle and the motivation of our approach in the response to the comment 6 in "notable observations and considerations".

*13/25: See "notable comments" on densimetric technique for high/hyperconcentrated streamflow SSC determinations.*

Answer: See the response to the comment 6 in "notable observations and considerations".

*15/3: Instead of "allows," perhaps "enables?"*

Answer: Yes, thanks. This has been corrected in the text as follows:" The device enables to measure the temporal evolution…"

*15/5+: Suspended solids vs SSC: I avoid the former unless the metric is Total Suspended Solids (TSS), which I studied 2 decades ago and declared fundamentally unreliable. See Gray et al. 2000 and other papers that "condemn" TSS data.*

Answer: thanks for the reference. Here the comment between SSC and TSS does not really apply because SCAF enables to measure a settling velocity distribution but not a particle size distribution. Nevertheless, we propose to replace "suspended solids" with "suspended sediment".

*15/8: SCAF is a very neat innovation. Congratulations. It in itself (with calibrating data) worth a paper?*

Answer: thanks. Yes, a paper will be published as soon as we have enough data including samples collected during floods.

*15/27: Echo-sounder is at-a-point…?*

Answer: Yes, as written on page/line 15/25, the echo-sounder is installed at a fixed point, usually a river bank or a bridge pier, with the sensor tilt with an angle of less than 25° from the vertical axis oriented towards the center of the river. It measures the evolution of the level of the river bed at a point of the cross section.

*17++ I neither reviewed "data transfer" nor "visualization" closely, as I wouldn't have much to add given you're far ahead of me on these (and certainly other) topic(s).*

Answer: OK, no comment.

*22/7: "Sites" or "gauges" on two rivers?*

Answer: OK, the text was changed as follows: "To date, RIPLE platform has been tested on ==two river sites located in the French Alps==…".

It was already specified in the text that the first site was previously gauged since there was an existing hydrometric station managed by the Electricité De France (EDF) company (22/12). On the other hand, it has been added that the second site was ungauged before the installation of RIPLE: "The second site corresponds to ==an ungauged site located in== a more pristine river in the Southern Alps where sediment loads can be high…"

*23/11: My Ph.D. dissertation was on "calibration coefficients for 4 pressure-difference bedload samplers." I am keenly interested in using surrogate bedload data, but first things first (in USA, anyway) is to make sure we know trapping efficiencies of our ground-truth instruments. I found that some pressure-difference bedload samplers had substantially super-efficient trapping coefficients.*

Answer: Trapping efficiency of pressure-difference bedload samplers is a very difficult question to raise. Determining the efficiency of a particular sampler deployed in a particular river is very difficult as the number of parameters influencing the measurements are high (bag clogging, position of sampler on a rough river-bed, size of sampler mouth versus bedload grain sizes, hydraulic efficiency…). It is indeed necessary to develop knowledge, share good practices on the use of these devices. However, the use of surrogate methods has to be considered as a step toward new data that are very difficult to obtain with sampling methods (such as continuous monitoring with remote transmission). Of course, validation of these surrogate methods have to be done. They will be continued by using "ground-truth" instruments.

*23/26: Dissolved Oxygen is (now) easily measured by luminescence technique, but of course its relevance is suspect-at-best in most non-tidal rivers.*

Answer: yes, that is a good idea. We could easily add such an oxygen sensor in the future to RIPLE.

*24/6: Aquatic drone. Sounds neat, but how to use? I'm a rapt observer…*

Answer: Images and videos illustrating the use of the aquatic drone are available on the website of the company that manufactures the drone: http://riverdrone.fr/.

Instruments such as echo-sounder, turbidimeter, aDcp, hydrophone can be embarked on the aquatic drone to obtain spatialized measurements and to carry out profiles across or along the river.

**References**

Bergougnoux L., Misguich-Ripault J., and Firpo J-L: Characterization of an optical fiber bundle sensor, Rev. Sci. Instrum. 69, 1985-1990, 1998.

Esteves, M., Legout, C., Navratil, O., and Evrard, O.: Medium term high frequency observation of discharges and suspended sediment in a Mediterranean mountainous catchment, Journal of Hydrology, 568, 562-574, https://doi.org/10.1016/j.jhydrol.2018.10.066, 2019.

Geay, T., Belleudy, P., Gervaise, C., Habersack, H., Aigner, J., Kreisler, A., Seitz, H. and Laronne, J. B.: Passive acoustic monitoring of bed load discharge in a large gravel bed river, J. Geophys. Res. Earth Surf., 21, https://doi.org/10.1002/2016JF004112, 2017a.

Geay, T., Belleudy, P., Laronne, J. B., Camenen, B., and Gervaise, C.: Spectral variations of underwater river sounds, Earth Surf. Proc. Land., 42, 2447–2456, https://doi.org/10.1002/esp.4208, 2017b.

Geay, T., L. Michel, S. Zanker, and J. R. Rigby (2019), Acoustic wave propagation in rivers: an experimental study, Earth Surf. Dyn., 7(2), 537–548, doi:10.5194/esurf-7-537-2019.

Lajeunesse, E., Delacourt, C., Allemand, P., Limare, A., Dessert, C., Ammann, J., Grandjean, P., Monitoring riverine sediment fluxes during floods : new tools and methods. https://hal.archives-ouvertes.fr/hal-01917118/document

Liébault, F., Jantzi, H., Klotz, S., Laronne, J.B., Recking, A., 2016. Bedload monitoring under conditions of ultra-high suspended sediment concentrations. J. Hydrol. 540, 947–958. https://doi.org/10.1016/j.jhydrol.2016.07.014.

Mano, V. : Processus fondamentaux conditionnant les apports de sédiments fins dans les retenues - optimisation des méthodes de mesure et modélisation statistique, Ph.D. thesis, Université Joseph-Fourier, 2008. Available at https://tel.archives-ouvertes.fr/tel-00365349 (tested on 02/10/2019).

Masihullah H. (2019). Estimation of flow discharge by non-contact measurements. Master 2, Hydraulics and Civil Engineering, ENSE3, G-INP.

Mathys, N. (2006). Analyse et modélisation à différentes échelles des mécanismes d'érosion et de transport de matériaux solides : cas des petits bassins versants de montagne sur marne (Draix, Alpes-de-Haute-Provence), PhD thesis, Université Joseph Fourier, France.

Navratil, O., Esteves, M., Legout, C., Gratiot, N., Némery, J., Willmore, S., and Grangeon, T.: Global uncertainty analysis of suspended sediment monitoring using turbidimeter in a small mountainous river catchment, J. Hydrol., 398, 246–259, https://doi.org/10.1016/j.jhydrol.2010.12.025, 2011.

Nord, G., Gallart, F., Gratiot, N., Soler, M., Reid, I., Vachtman, D., Latron, J., Martín Vide, J. P., and Laronne, J. B.: Applicability of acoustic Doppler devices for flow velocity measurements and discharge estimation in flows with sediment transport, J. Hydrol., 509, 504–518, https://doi.org/doi:10.1016/j.jhydrol.2013.11.020, 2014.

Nord, G., Boudevillain, B., Berne, A., Branger, F., Braud, I., Dramais, G., Gérard, S., Le Coz, J., Legoût, C., Molinié, G., Van Baelen, J., Vandervaere, J.-P., Andrieu, J., Aubert, C., Calianno, M., Delrieu, G., Grazioli, J., Hachani, S., Horner, I., Huza, J., Le Boursicaud, R., Raupach, T. H., Teuling, A. J., Uber, M., Vincendon, B., and Wijbrans, A. (2017) A high space–time resolution dataset linking meteorological forcing and hydro-sedimentary response in a mesoscale Mediterranean catchment (Auzon) of the Ardèche region, France, Earth Syst. Sci. Data, 9, 221-249, http://dx.doi.org/10.5194/essd-9-221-2017.

Singh, S.K., and Stenger, R.: Indirect Methods to Elucidate Water Flows and Contaminant Transfer Pathways through Meso-scale Catchments – a Review, Environ. Process. (2018) 5: 683. https://doi.org/10.1007/s40710-018-0331-6

Tafasca S. (2017). Use of velocity measurement to improve flow discharge estimation in a mountain stream. Master 2, Hydraulics and Civil Engineering, ENSE3, G-INP.

Thorne, P. D. (2014), An overview of underwater sound generated by interparticle collisions and its application to the measurements of coarse sediment bedload transport, Earth Surf. Dyn., 2(2), 531–543, doi:10.5194/esurf-2-531-2014.

P. D. Thorne, D. Hurther, An overview on the use of backscattered sound for measuring suspended particle size and concentration profiles in non-cohesive inorganic sediment transport, Continental Shelf Research, 73, pp. 97-118, 2014.

A. Vergne, C. Berni, J. Le Coz. Acoustic measurement of suspended sediments in rivers: potential impact of air micro-bubbles?. Underwater Acoustics Conference and Exhibition 2017, Sep 2017, Skiathos, Greece. pp.749-754. https://hal.archives-ouvertes.fr/hal-01768427/document

---

## Referee Comment (RC2) · Anonymous Referee #2 · 4 Jan 2020

The authors present a setup for a gauging station to monitor water and sediment transport in high temporal resolution. The station is designed for fast responding, mountainious, mesoscale rivers in remote areas. Both processes – water and sediment transport - are difficult to measure especially in these environments and subject to lots of efforts and uncertainties. Thus, the development of appropriate and robust monitoring techniques as well as their combined application is highly recommended and needed. This is also of special interest for process understanding and model parameterization regarding sediments. Several sensors are assembled in one monitoring system, measuring different parameters which serve as proxies or auxiliary variables to derive water and sediment discharge. The system also aims for data storage and

analysis (some measures of primary statistics), data transfer, and remote supervision and control. Also a user web-interface is provided. The manuscript is very good organized. The technical presentation of the RIPLE system with its several components is comprehensive and supported by meaningful figures and tables. However, a comprehensive presentation of the monitoring results is missing or only available partly. Here it would be good to give the reader an impression, what he/she could expect and how the (raw) measurements look like at least for one flood event. Sure, figure 8 shows some of them, but there are more graphs inside than explained and the legends are not self explaining. It would be beneficial if the authors show more results for the same event for other sensors/variables especially regarding the bedload. Figure 9 to 12 are less informative in this regard since they show other periods in time or aspects of minor interest like Figure 12. Furthermore, I have a few doubts about the English language. I am not a native speaker but some phrases sound a bit strange for me.

Here are some details:

1-19: You can shorten the abstract by removing the information in brackets.

1-23: "The RIPLE platform has been designed ... " - I don't understand the meaning of this sentences.

1-28/29: information in brackets not needed in the abstract.

3-4: split the sentence after "...the cross section"

3-6: Maybe, start sentence with "But this assumption ..."

3-9: without returned? → "... are analysed in the laboratory to measure ..."

3-23: aDcp or ADCP?

3-27: "record" instead of "consist in recording"

4-1: "Up to date, whatever the indirect method used, ... " → I think an english revision is required.

4-5: Suggest reversing the sentence: "Knowledge of . . ."

5-9: Suggest: "There are studies . . ." instead of

6-21: Suggest: "Therefore, they . . ." instead of "They therefore . . .

7-28: Suggest: "The camera is an AXIS P1427-E. Its selection based on various criteria such as image quality, . . .. "

8-25-32: You are using a standard method for deriving discharge from surface velocity measurements. You can also refer to USGS guidelines or ISO norms.

8-29: "depth averaged velocity" sounds a bit crazy since you measure only surface velocities

9-14: see comment 7-28

9-18: What is meant by "bitter points"? Do you mean "control point"?

9-25: What is meant by "sides of flow"? Do you mean "river bank"?

9-32: ". . . the interval . . . is constant . . ." instead of "be"

9-34: ". . .it is not taking movies." instead of "pictures" Or do you use single pictures from this camera too?

10-9: ". . .to illuminate the water surface." Without "up to"

10-11: "exceeded" instead of "overcome"

10-15: " . . . processing steps are executed in the laboratory . . ."

10-28: suggest "target area" instead of "area targeted"

10-34: suggest to remove ". . . with no connection to the data logger."

11-25: " . . . the radar derives the distance separating the radar from the water surface." Sentence is not understandable. Please revise. And what is its relation to the "blind

area" mentioned afterwards?

12-27: Please add a reference to "PASS".

13-10: "Above some . . . " ??? Suggest to rephrase: "If . . . thresholds are exceeded,

13-34: "display unit" instead of "Display Unit"

14-28: That's not possible since you write before that the senor is installed above low flow water levels.

15-1: Please avoid abbreviations in headlines.

16-2: suggest ". . . of angles less . . ." instead of ". . .with angles of less . . ."

16-5: 0.007m → Diameter of 7mm would be really nice. Is this right?

18-1: Suggest: "All procedures . . . are presented . . ."

19-18: Am I right that the ecco sounder is mounted next to the lowest water level looking towards the bottom?

21-19: Suggest: ". . . the FTP-server . . .." instead of "FTP"

22-20: You write ". . .the platform has worked properly, recording a large data set that will be of great interest for the understanding of sediment transport processes in alpine rivers." Please show an example of this data set at least by visualization of all measured variables for one flood event.

22-29: Please revise "much". Maybe: ". . . it is also adjustable and transferable ..."

23-15: Suggest: "These measurements must be performed under conditions that are close to the in situ environment in order to avoid subsequent flocculation/disaggregation processes."

24-8: You write ". . .of the same variables as those measured by RIPLE platform at other points . . ." How can bedload and suspended sediments be measured by drone? Please

revise this paragraph or remove it. You can also place your hints on improvements in remote sensing in the paragraphs before.

---

## Author Comment (AC2) · 27 Jan 2020

**Author comments to referee 2 comments on "An autonomous and low-power instrument platform for monitoring water and solid discharges in mesoscale rivers" by Nord et al.**

In the following, the reviewer comments appear in black italic and our answers are provided in blue. When there are quotations from the text of the article, they appear in quotation marks and the new or corrected parts are highlighted in yellow.

gi-2019-33-RC2.pdf

*The authors present a setup for a gauging station to monitor water and sediment transport in high temporal resolution. The station is designed for fast responding, mountainious, mesoscale rivers in remote areas. Both processes – water and sediment transport - are difficult to measure especially in these environments and subject to lots of efforts and uncertainties. Thus, the development of appropriate and robust monitoring techniques as well as their combined application is highly recommended and needed. This is also of special interest for process understanding and model parameterization regarding sediments. Several sensors are assembled in one monitoring system, measuring different parameters which serve as proxies or auxiliary variables to derive water and sediment discharge. The system also aims for data storage and analysis (some measures of primary statistics), data transfer, and remote supervision and control. Also a user web-interface is provided. The manuscript is very good organized. The technical presentation of the RIPLE system with its several components is comprehensive and supported by meaningful figures and tables. However, a comprehensive presentation of the monitoring results is missing or only available partly. Here it would be good to give the reader an impression, what he/she could expect and how the (raw) measurements look like at least for one flood event. Sure, figure 8 shows some of them, but there are more graphs inside than explained and the legends are not self explaining. It would be beneficial if the authors show more results for the same event for other sensors/variables especially regarding the bedload. Figure 9 to 12 are less informative in this regard since they show other periods in time or aspects of minor interest like Figure 12. Furthermore, I have a few doubts about the English language. I am not a native speaker but some phrases sound a bit strange for me.*

Answer: we sincerely thank referee #2 for his/her careful reading of the paper and the ways proposed to improve it. We have added a new figure (see below Figure 14) showing the raw data measured by 6 sensors for the event of 8 to 9 August 2017 in Bourg d'Oisans (Romanche river). The data are provided by the following sensors: water level radar, surface velocity radar, susix MJK turbidity sensor, conductivity probe, echo sounder and hydrophone. This figure complements Figure 9 of the paper which showed incomplete results since extracted from a snapshot of the user web-interface. This new figure is inserted at the end of the document in section 6 "Case study" (see below specific comment 22-20).

Additionally, for the period from 8 to 9 August 2017, the videos (provided by the Large Scale Particle Image Velocimetry digital camera), photos (provided by the control camera), the two text files "Riple_DATA.txt" (data from all instruments) and "Riple_SAV.txt" (data that enable to remotely control the proper functioning of the platform), and the raw data of the hydrophone are in open access via the following perennial link: https://doi.osug.fr/data/public/RIPLE_8to9August2017/

This provides a comprehensive picture of the data produced by the station in terms of formats, file sizes and data quality.

[Figure]

Figure 14: Raw data derived from the "Riple_DATA.txt" file for the 8 to 9 August 2017 in Bourg d'Oisans (Romanche river). (a) Time series of water level and surface water velocity. (b) Time series of water level and turbidity (susix MJK). Due to an inappropriate parameterization of the sensor, there is saturation of the signal above 1000 FNU. (c) Time series of water level and electrical

conductivity. (d) Time series of water level and temperature. (e) Time series of water level, distance from the echosounder to the river bed and acoustic power at 2 kHz measured by the hydrophone.

Regarding the remark about the English language, a large number of corrections proposed by referee #1 John Gray, who is a native English-speaker, as well as other corrections proposed by referee #2 have been taken into account.

*Here are some details:*

*1-19: You can shorten the abstract by removing the information in brackets.*

Answer: we would like to remain the information in brackets in the abstract as we believe it provides important clarification to readers, before eventually starting a more in-depth reading of the entire paper. We have only removed information on the geographical coordinates and elevations of the test sites.

*1-23: "The RIPLE platform has been designed… " - I don't understand the meaning of this sentences.*

Answer: we agree that this sentence is unclear and does not provide relevant information. We have deleted it.

*1-28/29: information in brackets not needed in the abstract.*

Answer: As mentioned above, we have removed this information from the abstract.

*3-4: split the sentence after "… the cross section"*

Answer: we agree, the sentences have been corrected as followed: "In turbulent rivers, it is assumed that the SSC is relatively homogeneous within the cross section. Therefore point measurement of turbidity from the river bank is acceptable."

*3-6: Maybe, start sentence with "But this assumption…"*

Answer: we agree, the sentences have been corrected as followed: "However this assumption is questionable for sand-sized particles (Camenen et al., 2019) and there is no reference method for sand-sized particles to date."

*3-9: without returned? -> "… are analysed in the laboratory to measure…"*

Answer: we agree, the sentences have been corrected and complemented as followed: "Samples collected at regular intervals or when thresholds are exceeded (e. g. water level, turbidity) are manually retrieved during site visits and brought to the laboratory. They are then analysed to measure…"

*3-23: aDcp or ADCP?*

Answer: as recommended also by referee #1, we have applied aDcp everywhere.

*3-27: "record" instead of "consist in recording"*

Answer: we agree, "consist in recording" has been replaced by "record".

*4-1: "Up to date, whatever the indirect method used,…" -> I think an english revision is required.*

Answer: we have read again the sentence and it does not seem problematic to us.

*4-5: Suggest reversing the sentence: "Knowledge of…"*

Answer: this correction was also proposed by referee #1 and it has been taken into account as followed: "Knowledge of water discharge is essential whether it is for estimating suspended sediment fluxes, dissolved matter fluxes, nutrient or contaminant fluxes associated with fine particles."

*5-9: Suggest: "There are studies …" instead of*

Answer: we agree, "There has been studies…" has been replaced by "There are studies…"

*6-21: Suggest: "Therefore, they…" instead of "They therefore…*

Answer: we agree, "They therefore …" has been replaced by "Therefore, they …"

*7-28: Suggest: "The camera is an AXIS P1427-E. Its selection based on various criteria such as image quality,…"*

Answer: we agree, the text has been corrected as followed: "The selected camera is an AXIS P1427-E. Its selection was based on various criteria…"

*8-25-32: You are using a standard method for deriving discharge from surface velocity measurements. You can also refer to USGS guidelines or ISO norms.*

Answer: we agree, thank you for this remark. The text has been complemented as followed: "A transect of surface velocity along the cross section of the river is extracted and converted to a transect of depth averaged velocity over the vertical using a coefficient that relates the depth averaged velocity to the surface velocity. Such coefficient commonly ranges between 0.75 and 0.85 (Hauet et al., 2008; Le Coz et al., 2010; Welber et al., 2016) but it is preferable to define it from aDcp or current meter measurements in accordance with ISO 748 and USGS guidelines."

The reference to the norm ISO 748 has also been added to the list of references:

"International Organization for Standardization, 2007: Hydrometry — Measurement of liquid flow in open channels using current-meters or floats. ISO 748:2007, Geneva."

*8-29: "depth averaged velocity" sounds a bit crazy since you measure only surface velocities*

Answer: yes, we understand your comment. It was also highlighted by referee #1. That is why we have reformulated the text as presented in the previous comment. We have added specifically the following recommendation " but it is preferable to define it from aDcp or current meter measurements in accordance with ISO 748 and USGS guidelines."

However, we should keep in mind that for steep rivers, with slope typically greater than 1%, and particularly in the case of mobile bed rivers, it is almost impossible to maintain conventional stage-discharge rating curves and to measure velocity distributions within the cross section during floods.

*9-14: see comment 7-28*

Answer: we agree, the text has been corrected as followed: "The selected camera is an AXIS P1435-LE. Its selection was based on various criteria…"

*9-18: What is meant by "bitter points"? Do you mean "control point"?*

Answer: yes, thank you. "bitter points" has been replaced by "control point" in the text.

*9-25: What is meant by "sides of flow"? Do you mean "river bank"?*

Answer: yes, thank you. "sides of flow" has been replaced by "river bank" in the text.

*9-32: "… the interval … is constant …" instead of "be"*

Answer: we have read again the sentence and it does not seem problematic to us.

*9-34: "… it is not taking movies." instead of "pictures" Or do you use single pictures from this camera too?*

Answer: yes, "pictures" has been replaced by "movies".

*10-9: "… to illuminate the water surface." Without "up to"*

Answer: we agree, "up to" has been removed from the text.

*10-11: "exceeded" instead of "overcome"*

Answer: yes, thank you. "overcome" has been replaced by "exceeded" in the text.

*10-15: "… processing steps are executed in the laboratory …"*

Answer: we agree, "will be" has been replaced by "are" in the text.

*10-28: suggest "target area" instead of "area targeted"*

Answer: we agree, "elliptical area targeted" has been replaced by "elliptical target area" in the text.

*10-34: suggest to remove "… with no connection to the data logger."*

Answer: yes, thank you. "with no connection to the data logger" has been removed.

*11-25: "… the radar derives the distance separating the radar from the water surface." Sentence is not understandable. Please revise. And what is its relation to the "blind area" mentioned afterwards?*

Answer: we agree, we have rewritten these two sentences as follows: "Accounting for this time of flight and the velocity of the wave in the air, which depends on the air temperature, the radar calculates the distance from the sensor to the water surface. Distances of less than 0.15 m cannot be measured, this is known as the blind zone."

*12-27: Please add a reference to "PASS".*

Answer: we agree, two references "(Huon et al., 2017; M-Tropics, 2017)" have been added in the text and in the list of references:

"Huon S., Evrard O., Gourdin E., Lefèvre I., Bariac T., Reyss J.L., des Tureaux T.H., Sengtaheuanghoung O., Ayrault S., and Ribolzi O.: Suspended sediment source and propagation during monsoon events across nested sub-catchments with contrasted land uses in Laos, J Hydrol: Regional Studies, 9, 69–84, 2017.

M-Tropics, Service de données OMP (SEDOO): Stations and acquisition parameters, [online] Available from: https://mtropics.obs-mip.fr/stations-and-acquisition-parameters/laos-lak-sip-catchment/?noredirect=en_US (Accessed 16 January 2020), 2017."

*13-10: "Above some … " ??? Suggest to rephrase: "If … thresholds are exceeded,*

Answer: we agree, "above some" has been replaced by "In case of exceeding" in the text.

*13-34: "display unit" instead of "Display Unit"*

Answer: yes, it has been corrected.

*14-28: That's not possible since you write before that the senor is installed above low flow water levels.*

Answer: this sentence was not clear. It has been rewritten as followed: "It was therefore decided to install it at a higher position than the Susix sensor so that it is immersed for a shorter period of time than the Susix sensor, i.e. generally only during the central part of the floods, around the peak of the flood."

*15-1: Please avoid abbreviations in headlines.*

Answer: we agree, "SCAF" has been replaced by "System characterizing aggregates and flocs" in the text.

*16-2: suggest "… of angles less …" instead of "… with angles of less …"*

Answer: we have decided to remain the initial formulation.

*16-5: 0.007m -> Diameter of 7mm would be really nice. Is this right?*

Answer: good remark. "0.007 m" has been replaced "0.07 m".

*18-1: Suggest: "All procedures … are presented …"*

Answer: we agree, the proposition has been applied.

*19-18: Am I right that the ecco sounder is mounted next to the lowest water level looking towards the bottom?*

Answer: not really. In the case of the Romanche river in Bourg d'Oisans, the echo-sounder is installed on the central pier of the bridge, about 1 m above the river bed looking down. The echo-sounder is inclined at an angle of about 20° from the vertical. As can be seen in Figure 9 and in the added figure (Figure 14), the distance from the echo-sounder to the river bed varies between 0.9 and 1.4 m during the flood event of the 8 to 9 August 2017.

*21-19: Suggest: "… the FTP-server …." instead of "FTP"*

Answer: we agree, "FTP" has been replaced by "FTP server"

*22-20: You write "… the platform has worked properly, recording a large data set that will be of great interest for the understanding of sediment transport processes in alpine rivers." Please show an example of this data set at least by visualization of all measured variables for one flood event.*

Answer: As presented above, a new Figure has been added. The end of section 6 was complemented as followed: "Figure 14 shows the raw data measured by 6 sensors for the event of 8 to 9 August 2017 on the Romanche river in Bourg d'Oisans. The data are provided by the following sensors: water level radar, surface velocity radar, Susix MJK turbidity sensor, conductivity probe, echo sounder and hydrophone. Figure 14 focuses on the same flood event as Figure 9 which was only a screenshot of the user web-interface. Additionally, the videos (provided by the Large Scale Particle Image Velocimetry digital camera), photos (provided by the control camera), the two text files "Riple_DATA.txt" (data from all instruments) and "Riple_SAV.txt" (data that enable to remotely control the proper functioning of the platform), and the raw data of the hydrophone are in open access for this flood event through this link: https://doi.osug.fr/data/public/RIPLE_8to9August2017/

This provides a comprehensive picture of the data produced by the station in terms of formats, file sizes and data quality."

*22-29: Please revise "much". Maybe: "… it is also adjustable and transferable …"*

Answer: we agree, the sentence has been corrected as followed: "The platform has been designed to be applied preferably to rivers in mountainous areas, but it is adjustable and transferable to lowland rivers."

*23-15: Suggest: "These measurements must be performed under conditions that are close to the in situ environment in order to avoid subsequent flocculation/disaggregation processes."*

Answer: we agree, "most closely resemble those of" has been replaced by "are close to".

*24-8: You write "… of the same variables as those measured by RIPLE platform at other points …" How can bedload and suspended sediments be measured by drone? Please revise this paragraph or remove it. You can also place your hints on improvements in remote sensing in the paragraphs before.*

Answer: we do not really understand this remark because the aquatic drone can carry submerged instruments to make measurements at any point in the river. Nevertheless, one sentence has been changed to make the things clearer: "The aquatic drone can thus carry submerged instruments (e.g. conductivity probe, turbidimeter, echo-sounder, hydrophone, automatic sampler) to perform measurement campaigns…"

A new sentence was also added concerning the point on remote sensing: "Remote sensing instruments (e.g. radiometers) could also be added to perform non-intrusive turbidity measurements."